# The *Salmonella* transmembrane effector SteD hijacks AP1-mediated vesicular trafficking for delivery to antigen-loading MHCII compartments

Camilla Godlee *, Ondrej Cerny¤, Mei Liu, Samkeliso Blundell, Alanna E. Gallagher, Meriam Shahin, David W. Holden*

MRC Centre for Molecular Bacteriology and Infection, Imperial College London, London, United Kingdom

¤ Current address: Institute of Microbiology of the Czech Academy of Sciences, Prague, Czech Republic
* c.godlee@imperial.ac.uk (CG); d.holden@imperial.ac.uk (DWH)

**Data Availability Statement:** All relevant data are within the manuscript and its Supporting Information files.

## Abstract

SteD is a transmembrane effector of the *Salmonella* SPI-2 type III secretion system that inhibits T cell activation by reducing the amounts of at least three proteins –major histocompatibility complex II (MHCII), CD86 and CD97 –from the surface of antigen-presenting cells. SteD specifically localises at the *trans*-Golgi network (TGN) and MHCII compartments; however, the targeting, membrane integration and trafficking of SteD are not understood. Using systematic mutagenesis, we identify distinct regions of SteD that are required for these processes. We show that SteD integrates into membranes of the ER/Golgi through a two-step mechanism of membrane recruitment from the cytoplasm followed by integration. SteD then migrates to and accumulates within the TGN. From here it hijacks the host adaptor protein (AP)1-mediated trafficking pathway from the TGN to MHCII compartments. AP1 binding and post-TGN trafficking require a short sequence in the N-terminal cytoplasmic tail of SteD that resembles the AP1-interacting dileucine sorting signal, but in inverted orientation, suggesting convergent evolution.

## Author summary

*Salmonella enterica* is an intracellular pathogen that causes a range of diseases from gastroenteritis to systemic typhoid fever. Its pathogenesis relies on virulence proteins known as effectors that are delivered into host cells and modulate host cellular processes. The ability of the *Salmonella* effector SteD to localise within host MHCII compartment membranes is essential for its function in disrupting the adaptive immune response. Here we show that SteD integrates into membranes of the early secretory pathway through a two-step recruitment and integration mechanism. SteD then behaves like a transmembrane cargo protein and hijacks a post-Golgi vesicular trafficking pathway to reach MHCII compartments. This study highlights the sophistication by which bacterial pathogens interact with host cell biology at the molecular level.

**Funding:** This work was supported by an Investigator Award from the Wellcome Trust (209411/Z/17/Z) to D.W.H. The funders had no role in study design, data collection and analysis, decision to publish, or preparation of the manuscript.

**Competing interests:** The authors have declared that no competing interests exist.

## Introduction

The virulence of many bacterial pathogens relies on the delivery of effector proteins into host cells through secretion systems such as the type three secretion system (T3SS). These effectors manipulate immune responses and promote bacterial replication. Many effectors require a specific host cellular localisation for their function [1,2]. A subset of bacterial effectors from diverse pathogens, localise by integrating into specific membranes of host cells. These include *Salmonella* SteD, SseF and SseG, *E. coli* Tir and *Chlamydia* Incs. These transmembrane effectors are often crucial to pathogenesis; however, their targeting, membrane integration and trafficking are poorly understood. It has been proposed that they could integrate into host membranes by either lateral transfer during translocation through the T3SS pore or direct integration following translocation into the cytoplasm [3].

Following uptake into a host cell, *Salmonella* resides within a membrane-bound compartment known as the *Salmonella*-containing vacuole (SCV), from which it delivers effectors of the *Salmonella* pathogenicity Island (SPI)-2 T3SS through the vacuolar membrane. One of these, SteD, reduces mature antigen-loaded major histocompatibility complex (mMHCII) and CD86 from the surface of infected antigen-presenting cells, resulting in a reduction in T cell activation [4]. It also reduces CD97 cell surface levels, which destablises immunological synapses formed between dendritic cells and T cells [5]. It thus has an inhibitory effect on the adaptive immune response to *Salmonella*. mMHCII and CD97 interact with SteD and are ubiquitinated by the NEDD4 family HECT E3 ubiquitin ligase WWP2, generating predominantly K63 linkages and resulting in their lysosomal degradation [5,6]. SteD is also ubiquitinated by WWP2 in a way that augments its activity yet results in its lysosomal degradation [6]. The mechanism underlying this activity involves an intramembrane interaction between SteD and the transmembrane protein TMEM127, which acts as an adaptor for WWP2 [6].

SteD is 111 amino acids in length, has two transmembrane domains and integrates into host cell membranes such that both the N and C termini are exposed to the cytoplasm, separated by a luminal loop. Interaction with the *Salmonella* chaperone SrcA is required for SteD solubility in the *Salmonella* cytoplasm and efficient translocation [7]. Following bacterial translocation or exogenous expression in host cells, the majority of SteD is at the *trans*-Golgi network (TGN) [4]. It also localises to endosomal compartments including MHCII compartments, where mMHCII and CD97 are also found [4,5].

Despite some understanding of the protein-protein interactions required for SteD function, it remains unclear how SteD integrates into membranes, what is required for its localisation, and whether its localisation at the TGN is important for function. Through mutagenesis, we have found three different regions of SteD that are required for its localization, integration, and vesicular trafficking. Our results suggest that following translocation, a cytoplasmic intermediate of SteD is recruited to the ER or Golgi, where it undergoes membrane integration before transport to the TGN. Through interaction with the TGN-associated adaptor protein (AP)1 complex SteD then co-opts a post-TGN vesicular trafficking pathway to MHCII compartments, where it carries out its function.

## Results

### Two regions of SteD are required for membrane integration

Ectopic expression of GFP-SteD in antigen-presenting cells recapitulates the bacterially translocated protein with respect to subcellular localisation, membrane integration and reduction of mMHCII cell surface levels [4]. This shows that these processes do not require any other *Salmonella* factor and that SteD function is not affected by the GFP tag. Therefore, we

investigated the requirements for membrane integration and localisation of SteD using this system. In previous work from our group, alanine scanning mutagenesis of sequential blocks of 5–7 amino acids resulted in 20 different mutants (S1A Fig), 18 of which localised correctly at the TGN [4]. The other two mutants (SteD$_{ala9}$ and SteD$_{ala13}$) were not detectable [4]. In Ste-D$_{ala9}$, the substituted residues (LMCLG) are in the N-terminal transmembrane domain (Fig 1A). In SteD$_{ala13}$, the substituted residues (SVSSG) are in the luminal loop (Fig 1A). In the presence of MG132 (an inhibitor of proteasome degradation), both mutants were detected by immunoblot (Fig 1B), indicating that mutation of either region results in protein degradation.

Next, we analysed whether these regions are important for SteD function by measuring cell surface levels of mMHCII by flow cytometry after expression of both mutants by transfection into Mel Juso cells in the presence of MG132. As expected, wild-type (wt) GFP-SteD decreased surface levels of mMHCII compared to untransfected cells (Figs 1C and S1B). For cells expressing GFP-SteD the presence of MG132 reduced but did not prevent the decrease in surface mMHCII (S1C Fig). However, in cells containing similar levels of either GFP-SteD$_{ala9}$, GFP-Ste-D$_{ala13}$ or GFP alone, there was no significant reduction in mMHCII surface levels (Figs 1C and S1B), demonstrating that the mutated regions are required for the function of SteD.

To test whether GFP-SteD$_{ala9}$ or GFP-SteD$_{ala13}$ integrate into mammalian cell membranes we subjected transfected cells to biochemical fractionation. Cell lysates were pelleted by ultra-centrifugation to distinguish cytoplasmic proteins (including actin) from membrane-associated and integral membrane proteins, and then pellets were solubilised with either urea (to extract peripheral membrane proteins, including Golgin-97) or RIPA buffer (to extract integral membrane proteins, including the DRα chain of MHCII). As expected, wt GFP-SteD, along with DRα, was present in the pellet after the initial centrifugation and the urea wash, but was solubilised by RIPA, confirming its integration into host membranes (Fig 1D). However, both GFP-SteD$_{ala9}$ and GFP-SteD$_{ala13}$ were resistant to solubilisation with RIPA (Fig 1D). This indicates that neither mutant underwent membrane integration, but instead formed insoluble aggregates. Therefore, amino acids within the mutated regions of SteD$_{ala9}$ (hereafter referred to as Region 9) and SteD$_{ala13}$ (hereafter referred to as Region 13) are required for membrane integration and functionality of SteD.

## A two-step process for post-translocation membrane integration

To determine how these mutations affected SteD localisation in the host cell, we examined GFP-SteD$_{ala9}$ and GFP-SteD$_{ala13}$ in the presence of MG132 by fluorescence microscopy. Whereas a large proportion of wt GFP-SteD colocalised with the TGN marker TGN46, GFP-SteD$_{ala9}$ formed cytoplasmic punctate structures that colocalised with ubiquitin (Figs 2A and S2A). This, along with its resistance to detergent extraction (Fig 1D) indicates that when proteasome activity is inhibited, GFP-SteD$_{ala9}$ forms cytoplasmic aggregates that resemble aggresomes or inclusion bodies [8]. In contrast, SteD$_{ala13}$ colocalised partially with ubiquitin (S2A Fig) and predominantly with TGN46, although to a lesser extent than wt GFP-SteD (Fig 2A and 2B). Time-lapse microscopy revealed that wt GFP-SteD was present in motile vesicles, whereas GFP-SteD$_{ala13}$ remained stably associated with the Golgi area (S1 Video). Therefore, in the presence of MG132, SteD$_{ala13}$ remained in a Golgi-associated non-integrating state, which then formed insoluble aggregates after cell lysis (Fig 1D). These results show that SteD localisation at the Golgi can be uncoupled from membrane integration, indicating that Region 9 mediates interaction with a membrane component, while Region 13 is required for integration following Golgi association.

To test whether Region 13 promotes membrane integration of other transmembrane sequences, we created a chimeric protein in which the transmembrane domains of SteD were

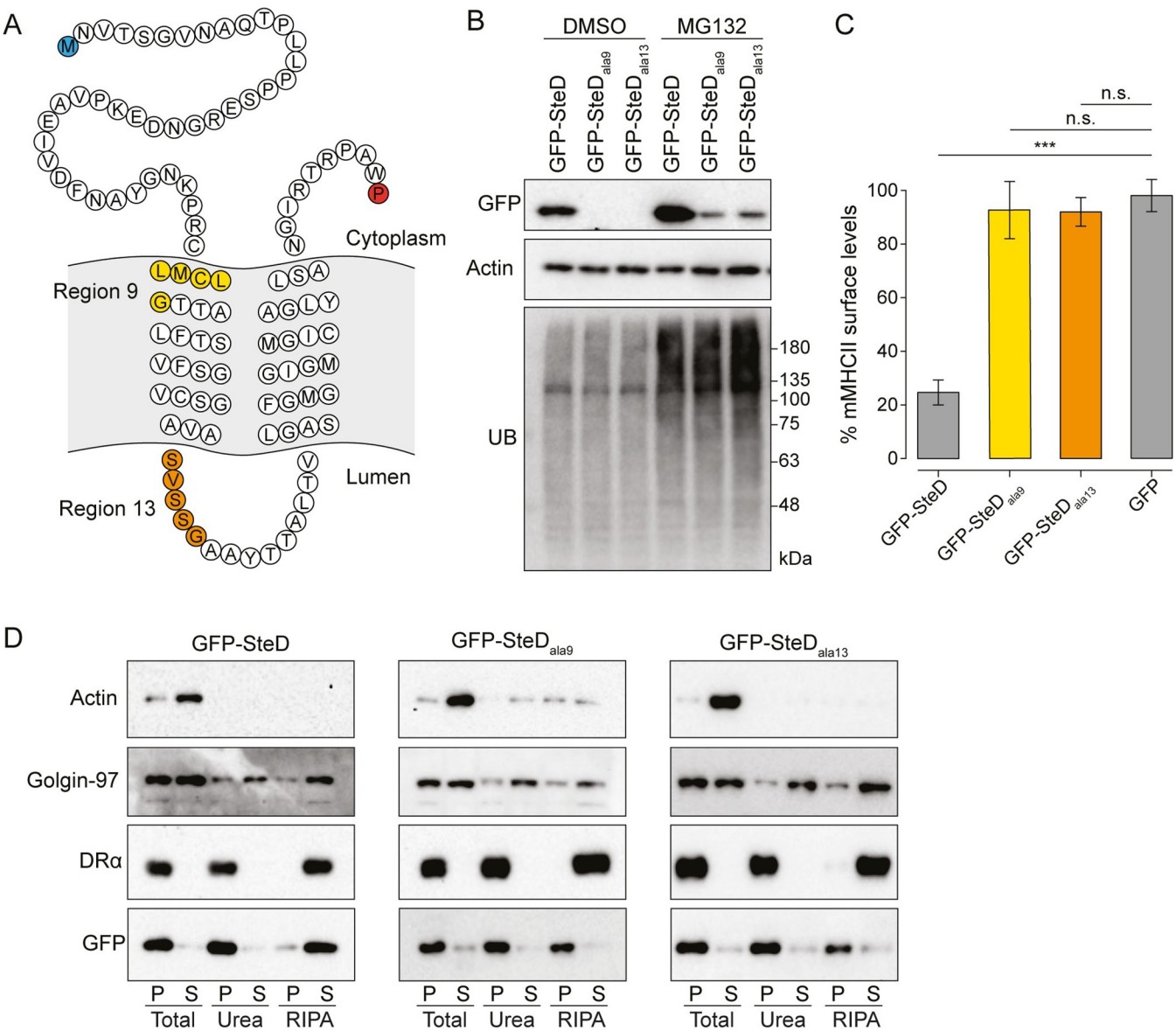

**Fig 1. Two regions of SteD are required for membrane integration.** (A) Amino acid sequence of SteD showing predicted transmembrane domains. The N-and C-terminal residues are highlighted in blue and red respectively. The residues substituted to alanines in SteD$_{ala9}$ and SteD$_{ala13}$ mutants are highlighted in yellow and orange respectively. (B) Protein immunoblots of whole-cell lysates derived from Mel JuSo cells expressing GFP-SteD (wt or mutants) and treated with MG132 or DMSO carrier. UB–ubiquitin. (C) mMHCII surface levels of Mel JuSo cells expressing GFP or GFP-SteD (wt or mutants) and treated with MG132. Cells were analysed by flow cytometry and amounts of surface mMHCII in GFP-positive cells are expressed as a percentage of GFP-negative cells in the same sample. Mean of three independent experiments done in duplicate ± SD. Data were analysed by one-way ANOVA followed by Dunnett's multiple comparison test, *** p<0.001, n.s.–not significant. (D) Protein immunoblots of membrane fractionation samples from Mel Juso cells expressing GFP-SteD (wt or mutants) and treated with MG132. Samples were taken from the pellet (P) and supernatant (S) of the total sample, after urea wash and after RIPA wash.

replaced by those of another *Salmonella* integral membrane effector–SseG, which has a similar membrane topology to SteD. This construct (SteD$_{TMSseG}$, Fig 2C) therefore contained SteD Region 13 but lacked Region 9. Fractionation experiments demonstrated that SteD$_{TMSseG}$ underwent membrane integration (Fig 2D), and this was dependent on Region 13, as demonstrated by the lack of RIPA solubility following its alanine substitution (SteD$_{TMSseGala13}$, Fig 2D). In contrast to SteD, SseG localised at membranes throughout the cytoplasm (Fig 2E).

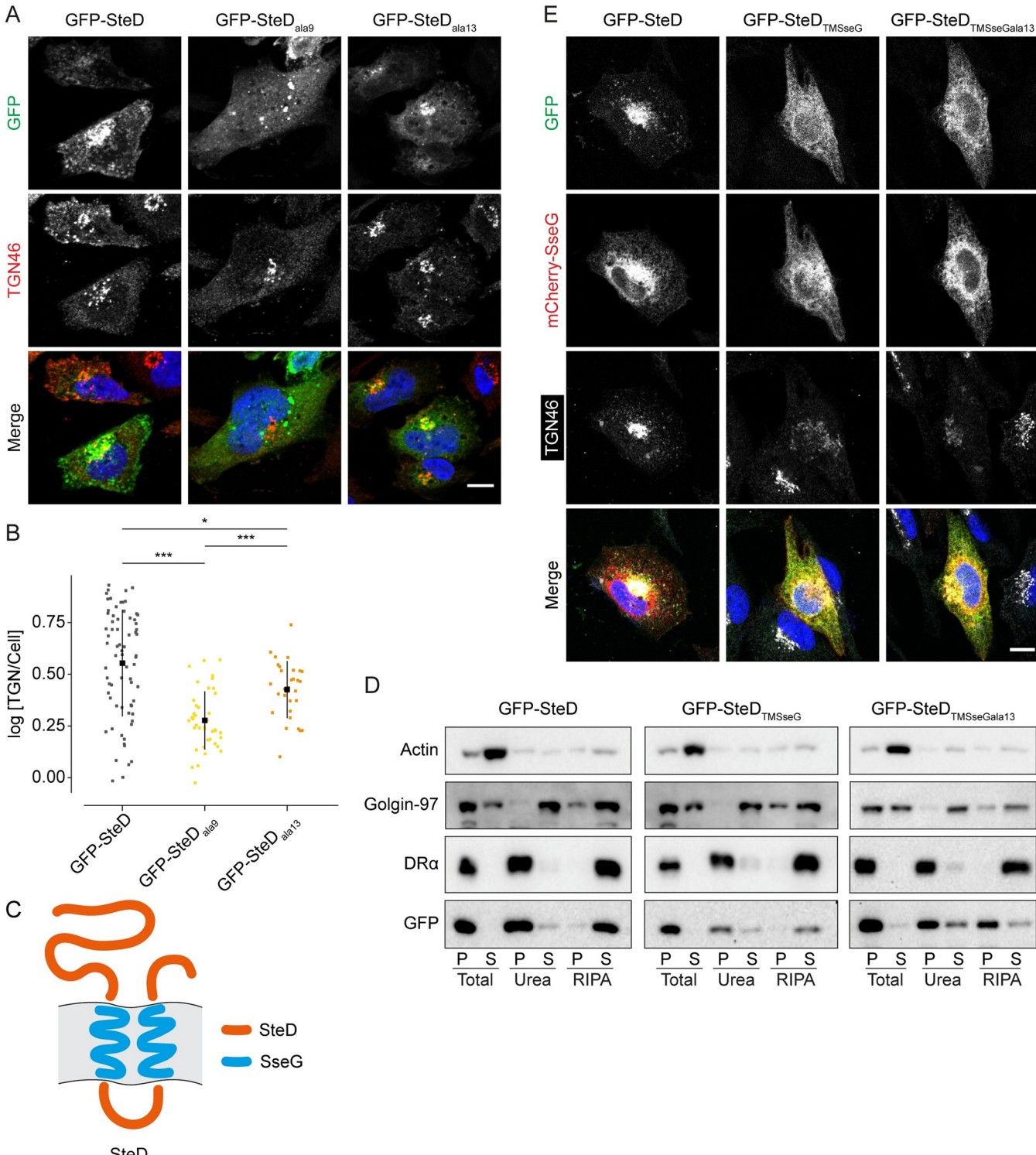

**Fig 2. Recruitment of SteD to the TGN is independent from membrane integration.** (A) Representative confocal immunofluorescence microscopy images of Mel JuSo cells expressing GFP-SteD (wt or mutants) after MG132 treatment. Cells were fixed and processed for immunofluorescence microscopy by labelling for the TGN (TGN46, red), and DNA (DAPI, blue). Scale bar– 10 μm. (B) Quantification of GFP at the TGN of cells represented in Fig 2A. The fluorescence intensity of the GFP signal at the TGN was measured in relation to total cellular fluorescence. Data are representative of three independent experiments. Each dot represents the value for one cell. Mean ± SD. The $\log_{10}$ fold change of the data were analysed by one-way ANOVA followed by Dunnett's multiple comparison test, *** $p < 0.001$, * $p < 0.05$. (C) Schematic of SteD chimera with transmembrane domains of SseG. (D) Protein immunoblots of

membrane fractionation samples from Mel Juso cells expressing GFP-SteD (wt or chimeric mutants) and treated with MG132. Samples were taken from the pellet (P) and supernatant (S) of the total sample, after urea wash and after RIPA wash. (E) Representative confocal immunofluorescence microscopy images of Mel JuSo cells expressing GFP-SteD (wt or chimeric mutants) and mCherry-SseG after MG132 treatment. Cells were fixed and processed for immunofluorescence microscopy by labelling for the TGN (TGN46, grey), and DNA (DAPI, blue). Scale bar– 10 μm.

Both GFP-SteD$_{TMSseG}$ and GFP-SteD$_{TMSseGGala13}$ had a similar localisation to mCherry-SseG (Fig 2E). This agrees with previous work showing that the transmembrane domains of SseG direct its localisation in the host cell [9] and demonstrates that this localisation is also independent from membrane integration. Therefore, Region 13 is sufficient to mediate integration of alternative transmembrane domains following targeting to a different membrane compartment.

We next examined the process of membrane integration after translocation of SteD by the SPI-2 T3SS. However, SteD$_{ala9}$ and SteD$_{ala13}$ were not stably expressed or translocated by *Salmonella* (S3A Fig). Therefore, we carried out more specific mutagenesis to identify the residues within Regions 9 and 13, whose loss accounts for the properties of SteD$_{ala9}$ and SteD$_{ala13}$. To do this we first tested the ability of GFP-tagged single and double residue alanine substitutions to reduce mMHCII surface levels. All the single residue substitutions were still functional (S3B Fig). The double substitution mutants with the strongest functional impairment (SteD$_{L42A, M43A}$ and SteD$_{S68A,G69A}$, Figs 3A and S3C) were tested for expression in *Salmonella*. Both mutants were translocated into Mel Juso cells and the protein levels were rescued by MG132, demonstrating that they both underwent proteasomal degradation (S3D Fig). Mel Juso cells with similar levels of SteD$_{L42A,M43A}$-HA and SteD$_{S68A,G69A}$-HA as wt SteD-HA had no reduction in mMHCII surface levels (Figs 3B and S3E). Biochemical fractionation revealed that both SteD$_{L42A,M43A}$-HA and SteD$_{S68A,G69A}$-HA were not solubilised by RIPA indicating that they failed to integrate into host cell membranes (Fig 3C). Furthermore, SteD$_{L42A,M43A}$-HA was found in large puncta throughout the host cell cytoplasm, while SteD$_{S68A,G69A}$-HA colocalised with TGN46 (Fig 3D and 3E). Therefore, following translocation, these mutants recapitulated the non-integration and localisation properties of GFP-SteD$_{ala9}$ and GFP-SteD$_{ala13}$ respectively, indicating that translocated SteD uses the same integration mechanism as ectopically expressed GFP-SteD.

Collectively, these results show that following translocation from bacteria, SteD integrates into host cell membranes with a two-step mechanism. We propose that SteD is first recruited from the cytoplasm by residues L42 and M43 of Region 9, presumably by interaction with a membrane component(s) and this is followed by integration mediated by residues S68 and G69 of Region 13.

## SteD integrates into membranes of the early secretory pathway

We investigated whether the membrane interaction and integration steps occur at the TGN or if SteD accumulates there after integrating elsewhere in the secretory pathway. To do this we used a doxycycline-regulated promoter to derepress expression of SteD from a mammalian expression plasmid after blocking Golgi trafficking with brefeldin A (BFA). BFA causes the collapse of Golgi membranes into the ER and fusion of TGN membranes with endosomes, thereby separating the early from the late secretory pathway [10,11]. When SteD expression was induced in the absence of BFA (dox), it integrated into membranes at both the TGN and MHCII compartments, as expected (Figs 4A–4C and S4A). When expressed after Golgi disruption (BFA-dox) SteD still underwent membrane integration (Fig 4C) but localised predominantly at the ER and no longer colocalised with TGN46 (Fig 4A and 4B). GFP-SteD$_{ala13}$ also localised at the ER when expressed after BFA treatment in the presence of MG132 (Fig 4A and 4B, BFA-dox),

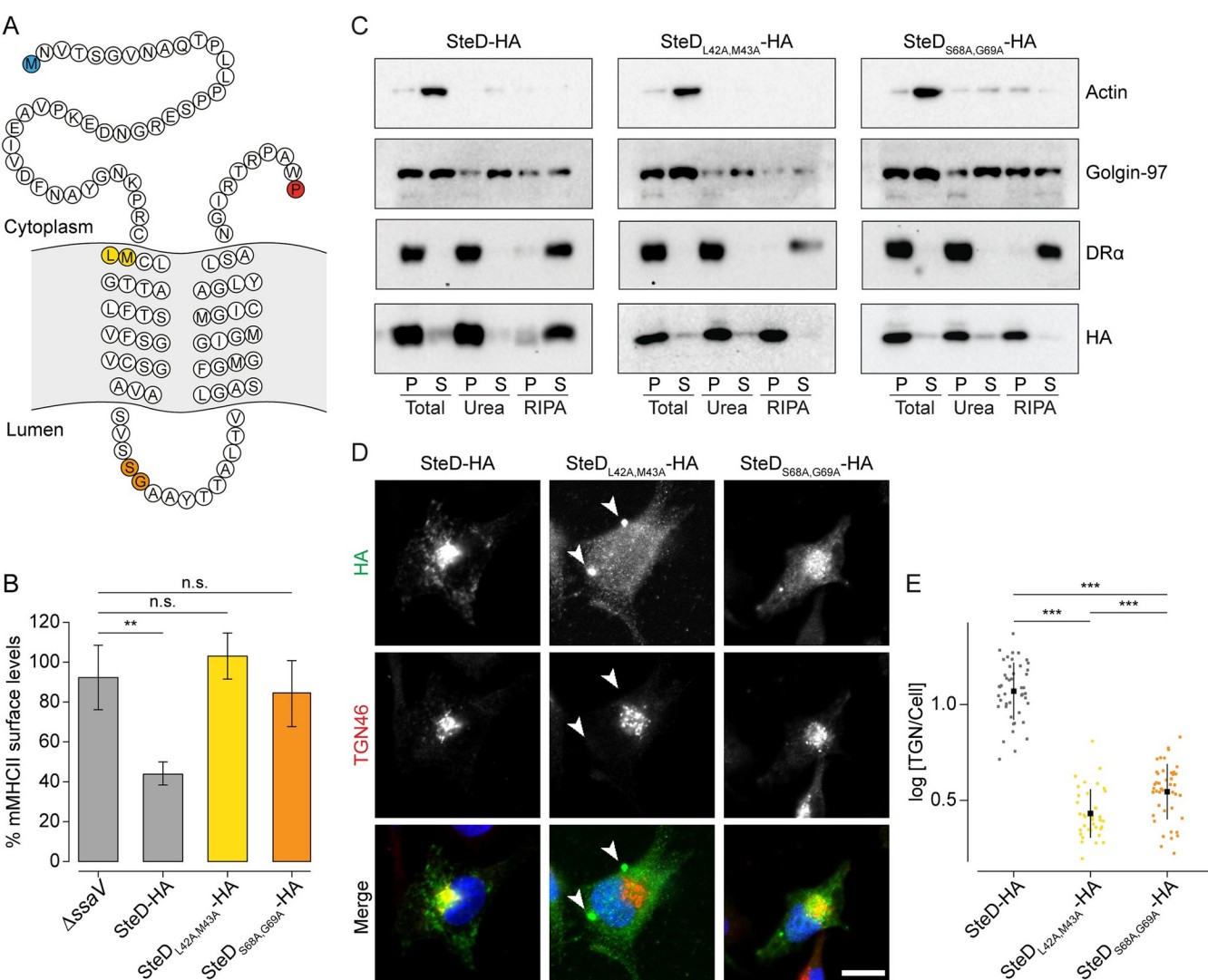

**Fig 3. Membrane integration following translocation.** (A) Amino acid sequence of SteD showing predicted transmembrane domains. The N- and C-terminal residues are highlighted in blue and red respectively. The residues substituted to alanines in SteD$_{L42A,M43A}$ and SteD$_{S68A,G69A}$ mutants are highlighted in yellow and orange respectively. (B) mMHCII surface levels of Mel JuSo cells infected with *ΔsteD Salmonella* carrying a plasmid expressing SteD-HA (wt or mutants) or SPI-2 null *ΔssaV Salmonella* and treated with MG132. Cells were analysed by flow cytometry and amounts of surface mMHCII in HA-positive cells are expressed as a percentage of HA-negative cells in the same sample. Mean of three independent experiments done in duplicate ± SD. Data were analysed by one-way ANOVA followed by Dunnett's multiple comparison test, ** p<0.01, n.s.–not significant. (C) Protein immunoblots of membrane fractionation samples from Mel Juso cells infected with *ΔsteD Salmonella* strains carrying a plasmid expressing SteD-HA (wt or mutants) and treated with MG132. Samples were taken from the pellet (P) and supernatant (S) of the total sample, after urea wash and after RIPA wash. (D) Representative confocal immunofluorescence microscopy images of Mel Juso cells infected with *ΔsteD Salmonella* strains carrying a plasmid expressing SteD-HA (wt or mutants) and treated with MG132. Cells were fixed and processed for immunofluorescence microscopy by labelling for HA (green), the TGN (TGN46, red), and DNA (DAPI, blue). Arrowheads indicate cellular aggregates. Scale bar– 10 μm. (E) Quantification of HA signal at the TGN of cells represented in Fig 3D. The fluorescence intensity of the HA signal at the TGN was measured in relation to total cellular fluorescence. Data are representative of three independent experiments. Each dot represents the value for one cell. Mean ± SD. The log$_{10}$ fold change of the data were analysed by one-way ANOVA followed by Dunnett's multiple comparison test, *** p<0.001.

demonstrating that the SteD membrane interaction partner re-distributes to the ER after BFA treatment. Therefore, both membrane interaction and integration occur at the ER or Golgi, and not at the TGN.

In the presence of BFA, SteD did not cause a significant reduction in mMHCII surface levels (Figs 4D and S4B) and this correlated with a failure to colocalise with mMHCII (S4A Fig).

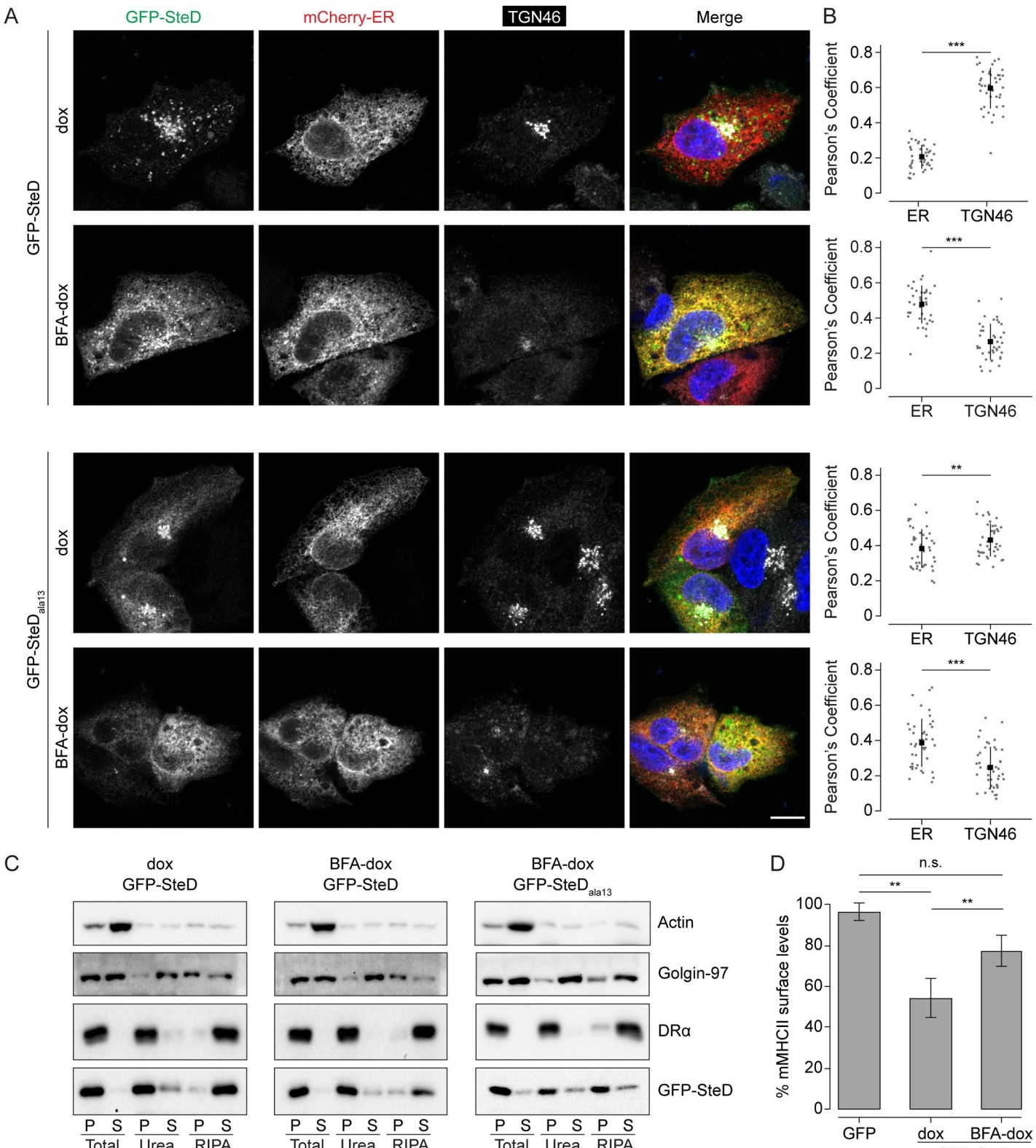

**Fig 4. SteD integrates into membranes of the early secretory pathway.** (A) Representative confocal immunofluorescence microscopy images of Mel JuSo cells expressing GFP-SteD (wt or mutant) under a doxycycline-regulated promoter and the ER marker mCherry-ER-3 (red). Cells were either treated with doxycycline plus MG132 (dox) or treated with BFA followed by doxycycline, MG132 and BFA (BFA-dox). Cells were then fixed and processed for immunofluorescence microscopy by

labelling for the TGN (TGN46, grey), and DNA (DAPI, blue). Scale bar– 10 μm. (B) Quantification of cells represented in Fig 4A. Pearson's correlation coefficients for colocalization between GFP-SteD and mCherry-ER-3 or TGN46. Data are representative of three independent experiments. Each dot represents the value for one cell. Mean ± SD. Data were analysed by paired t-test *** $p < 0.001$, ** $p < 0.01$. (C) Protein immunoblots of membrane fractionation samples from Mel Juso cells expressing GFP-SteD (wt or mutant) under a doxycycline-regulated promoter and treated as in Fig 4A. Samples were taken from the pellet (P) and supernatant (S) of the total sample, after urea wash and after RIPA wash. (D) mMHCII surface levels of Mel JuSo cells expressing GFP-SteD under a doxycycline-regulated promoter or GFP and treated as in Fig 4A. Cells were analysed by flow cytometry and amounts of surface mMHCII in GFP-positive cells are expressed as a percentage of GFP-negative cells in the same sample. Mean of three independent experiments done in duplicate ± SD. Data were analysed by one-way ANOVA followed by Dunnett's multiple comparison test, ** $p < 0.01$, n.s.–not significant.

This demonstrates that post-Golgi trafficking is required for SteD to reach MHCII compartments and for SteD function. Indeed, after bleaching the fluorescent signal from GFP-SteD outside of the Golgi area we detected vesicles containing GFP-SteD budding from the Golgi/TGN area by time-lapse microscopy. These vesicles trafficked throughout the cell with some apparently fusing to and budding from other SteD-containing compartments (S2 Video and S4C Fig). Since there is a large amount of colocalisation between GFP-SteD and mMHCII in the periphery of the cell [4], a substantial proportion of these are likely to be MHCII compartments. The stable association of SteD$_{ala13}$ with the TGN (Fig 2A and S1 Video) indicates that post-TGN trafficking of SteD requires membrane integration.

Therefore, rather than integrating into the TGN directly, SteD either interacts with an ER or Golgi cisternae component and integrates into these membranes before accumulating at the TGN. SteD is then trafficked to MHCII compartments, and this is required for its function.

## AP1 mediates trafficking of SteD to MHCII compartments via a sequence resembling an inverted dileucine motif

The TGN acts as a sorting platform for the anterograde traffic of protein cargo from the Golgi. Transmembrane cargo proteins frequently use short linear motifs known as sorting signals in their cytoplasmic tails. Sorting signals are recognised by cytoplasmic adaptor proteins, including heterotetrameric AP complexes, allowing concentration of cargo into vesicles targeted to specific membrane compartments [12]. In the absence of a sorting signal, proteins traffic constitutively from the TGN to the PM [13]. To determine if SteD interacts with an AP complex we used coimmunoprecipitation experiments after chemical crosslinking with dithiobis(succinimidyl propionate) (DSP), which has been shown to be effective for detecting transient interactions between transmembrane proteins and cytoplasmic interaction partners [14]. Ectopically expressed GFP-SteD was immunoprecipitated from Mel Juso cell lysates using GFP-trap beads and proteins were subjected to immunoblotting with antibodies against specific subunits of the AP1, 2 or 3 complexes. GFP-tagged SseG was used as a negative control. A small amount of AP1 interacted specifically and reproducibly with GFP-SteD (Fig 5A and 5B) but not GFP-SseG.

AP1 regulates the trafficking of protein cargo between the TGN and endosomes [15]. To test whether AP1 is involved in trafficking of SteD to MHCII compartments we used siRNA to knock down the β subunit of AP1 in Mel Juso cells expressing GFP-tagged SteD (Fig 5C). AP1 knockdown had no noticeable effect on cellular mMHCII signal but caused a significant reduction in colocalisation between GFP-SteD and MHCII compartments when compared to mock-treated cells or knockdown of AP2 and AP3 (Fig 5D and 5E). Despite this reduced colocalisation, knockdown of AP1 had no detectable effect on SteD-dependent reduction of mMHCII surface levels (S5A Fig). While there was no detectable increase in GFP-SteD at the PM (S5B Fig) we conclude that under these conditions a small proportion of SteD is still able to interact with MHCII either intracellularly or at the PM.

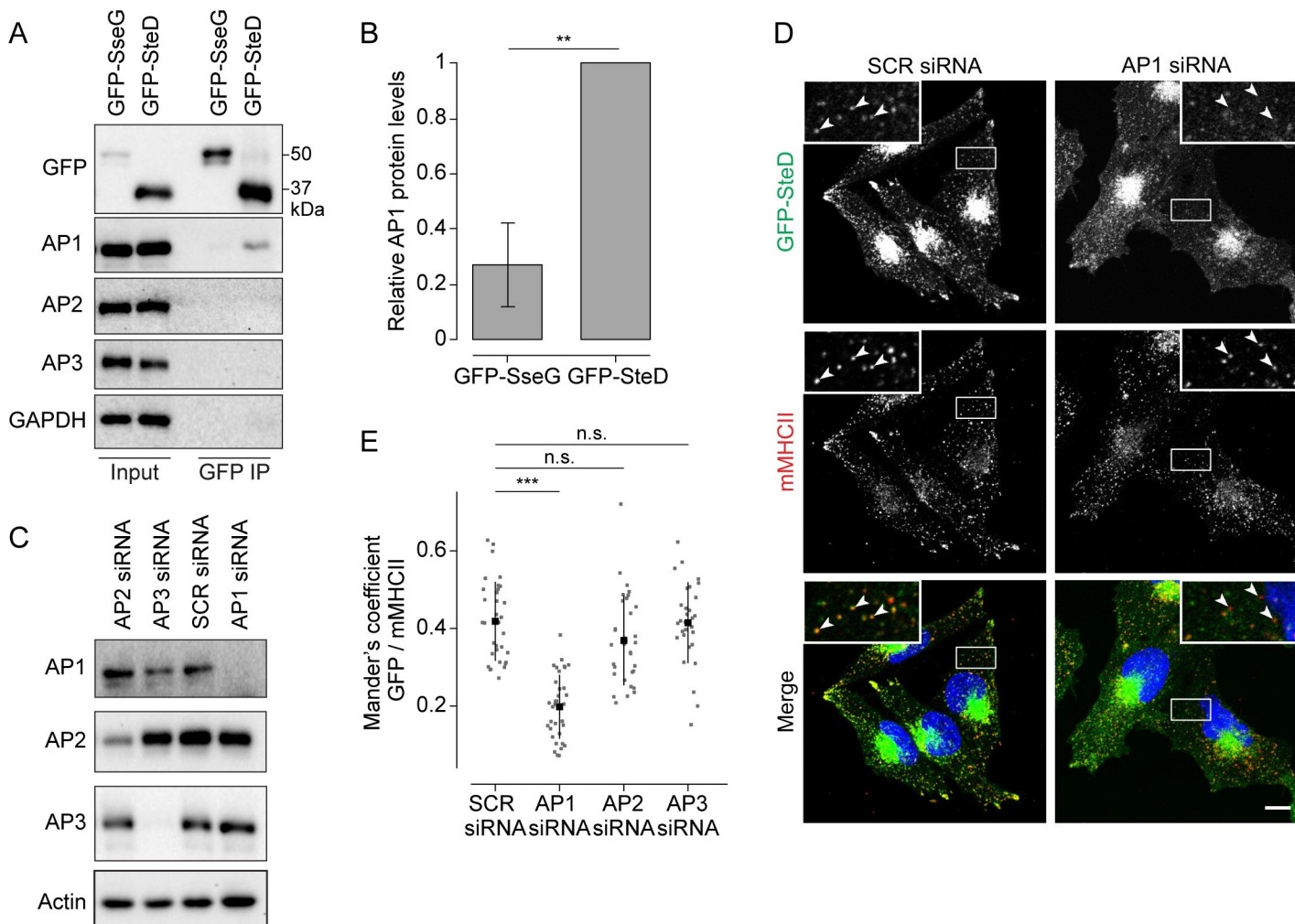

**Fig 5. AP1 mediates post-TGN trafficking of SteD.** (A) Protein immunoblots of whole-cell lysates (Input) and immunoprecipitation with GFP-trap beads (GFP IP) from Mel Juso cells expressing GFP-SteD or GFP-SseG following crosslinking with DSP. AP1 –antibody specific for the γ subunit, AP2 –antibody specific for the α subunit, AP3 –antibody specific for the δ subunit. (B) Levels of immunoprecipitated AP1 were calculated by densitometry from immunoblots as represented in Fig 5A. Protein levels were normalised to GFP-SteD. Mean of three independent experiments ± SD. The data were analysed by one sample t-test, ** p<0.01. (C) Protein immunoblots of Mel JuSo cells treated with scrambled siRNA (SCR) or siRNA specific to the β subunit of AP1, the μ subunit of AP2 or the δ subunit of AP3. AP1 – antibody specific for the β subunit, AP2 –antibody specific for the α subunit AP3 –antibody specific for the δ subunit. (D) Representative confocal immunofluorescence microscopy images of Mel JuSo cells expressing GFP-SteD after treatment with scrambled siRNA (SCR) or siRNA specific to the β subunit of AP1. Cells were fixed and processed for immunofluorescence microscopy by labelling for MHCII compartments (mMHCII, red), and DNA (DAPI, blue). Arrowheads indicate MHCII compartments. Scale bar– 10 μm. (E) Mander's overlap coefficient of the fraction of GFP-SteD positive pixels that colocalise with mMHCII positive pixels from cells after treatment with siRNA as in Fig 5C and 5D. Data are representative of three independent experiments. Each dot represents the value for one cell. Mean ± SD. Data were analysed by one-way ANOVA followed by Dunnett's multiple comparison test, *** p<0.001, n.s.–not significant.

We next tested whether there is information in the N- or C-terminal cytoplasmic tails of SteD that is required for its AP1-mediated trafficking. Truncation mutants of GFP-SteD lacking the C-terminal tail or most of the N-terminal tail (Figs 6A and S6A) underwent membrane integration as assessed by biochemical fractionation (Fig 6B). Truncation of the C-terminal tail (GFP-SteD$_{1-102}$) did not affect localisation at the TGN and MHCII compartments (Figs 6C and S6B). On the other hand, an SteD truncation lacking most of the N-terminal cytoplasmic tail (GFP-SteD$_{37-111}$) had reduced localisation at MHCII compartments (Fig 6C). Furthermore, and in contrast to AP1 depletion, truncation of the N-terminal tail resulted in a dramatic

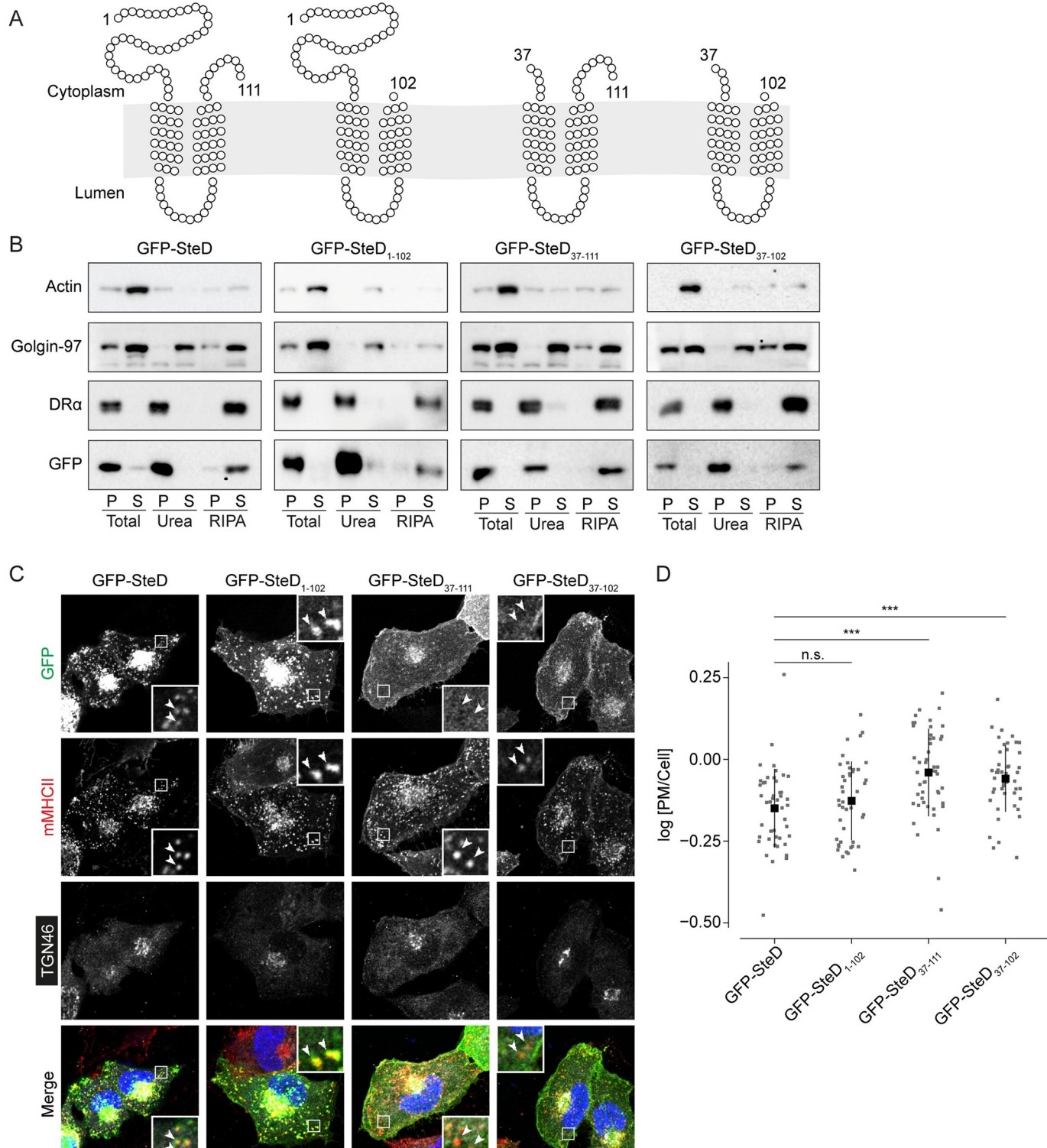

**Fig 6. The N-terminal tail of SteD is required for trafficking to MHCII compartments.** (A) Schematics of SteD showing predicted transmembrane domains and extent of truncation mutations as indicated. (B) Protein immunoblots of membrane fractionation samples from Mel Juso cells expressing GFP-SteD (wt or mutants). Samples were taken from the pellet (P) and supernatant (S) of the total sample, after urea wash and after RIPA wash. (C) Representative confocal immunofluorescence microscopy images of Mel JuSo cells expressing GFP-SteD (wt or mutants). Cells were fixed and processed for immunofluorescence microscopy by labelling for MHCII compartments (mMHCII, red), the TGN (TGN46, grey), and DNA (DAPI, blue). Arrowheads indicate MHCII compartments.

Scale bar– 10 μm. (D) Quantification of GFP at the surface of cells represented in Fig 6C. The fluorescence intensity of the GFP signal at the surface of cells was measured in relation to total cellular fluorescence. Data are representative of three independent experiments. Each dot represents the value for one cell. Mean ± SD. The $\log_{10}$ fold change of the data were analysed by one-way ANOVA followed by Dunnett's multiple comparison test, *** $p < 0.001$, n.s.–not significant.

increase in the proportion of fluorescence signal at the cell surface and a decrease in fluorescence signal at the TGN (Figs 6C and 6D and S6B). The continuous distribution of fluorescence signal at the cell surface along with the ability to integrate into the membrane implies a PM localisation. Remarkably, a construct lacking most of the N-terminal tail and the C-terminal tail (GFP-SteD$_{37-102}$), leaving just 65 residues comprising the two transmembrane domains separated by the luminal loop and containing Regions 9 and 13, also integrated efficiently into host cell membranes (Fig 6A and 6B) and along with GFP-SteD$_{37-111}$, localised at the TGN as well as the PM (Fig 6C and 6D).

To test whether the PM localisation was due to traffic from the TGN, we used the photoconvertible fluorescent protein mEos, which converts from green to red fluorescence upon activation with UV light, to specifically activate SteD at the Golgi and track its subsequent fate. As expected, vesicles containing mEos-SteD travelled from the Golgi throughout the cell cytoplasm (S3 Video and S6C Fig). In contrast, vesicles containing the N-terminal truncation of SteD trafficked to the periphery of the cell, where the fluorescent signal dissipated (S3 Video and S6C Fig), presumably as a result of vesicle fusion and lateral dilution of the fluorescent signal in the PM. These results suggest the presence of a sorting signal in the N-terminal tail of SteD that directs traffic from the TGN to MHCII compartments and whose absence results in mis-trafficking to the PM.

There are two well-characterised sorting signals that interact with AP complexes: the tyrosine motif, YXXΦ, where Φ is a large hydrophobic residue, and the dileucine motif, [DE]XXXL[LI], with one or more acidic residues upstream from two leucines (X indicates any residue in both motifs) [16]. No series of residues in SteD match the consensus of either motif, however sequences resembling both motifs are present in inverted orientations in the N terminal tail. F32, N33, A34 and Y35 resemble an inverted tyrosine motif, and are within a region necessary for SteD function, as determined by alanine scanning mutagenesis (Region 7, S1A Fig) [4]. A double alanine substitution of F32 and Y35 (SteD$_{F32A,Y35A}$) was sufficient to inhibit the effect of SteD on mMHCII surface levels to the same level as SteD$_{ala7}$ following ectopic expression or translocation from *Salmonella* (S7A and S7B Fig). Alanine substitution of the remaining residues within Region 7, N33 and G36, had no effect on SteD function (S7A Fig). However, the F32A, Y35A double mutation had no noticeable effect on the localisation of SteD (S7C–S7E Fig) and did not prevent interaction of SteD with AP1 when expressed ectopically (S7F and S7G Fig). This rules out the involvement of these residues in AP1-dependent transport and suggests that they contribute to SteD function in other ways.

Amino acids L13 and L14 of SteD, followed by P, P, S and then by two charged residues (E18, R19), resemble an inverted dileucine motif (Fig 7A). Alanine substitution of the leucines alone (SteD$_{L13A,L14A}$-HA) had no noticeable effect on localisation of SteD after translocation from *Salmonella* (S7H Fig). However, further alanine substitution of the two charged residues (E18 and R19) (SteD$_{L13A,L14A,E18A,R19A}$-HA) significantly increased the proportion of HA signal at the cell surface (Fig 7B and 7C) and reduced the level of colocalisation with mMHCII (Fig 7B and 7D) when compared to wt SteD-HA. An increase in cell surface signal and decrease in colocalisation with mMHCII was also detected for ectopically expressed GFP-SteD$_{L13A,L14A,E18A,R19A}$ compared to wt GFP-SteD (S7C–S7E Fig). Therefore, this quadruple substitution mutation recapitulated the mis-localisation phenotype of the N-terminal truncation

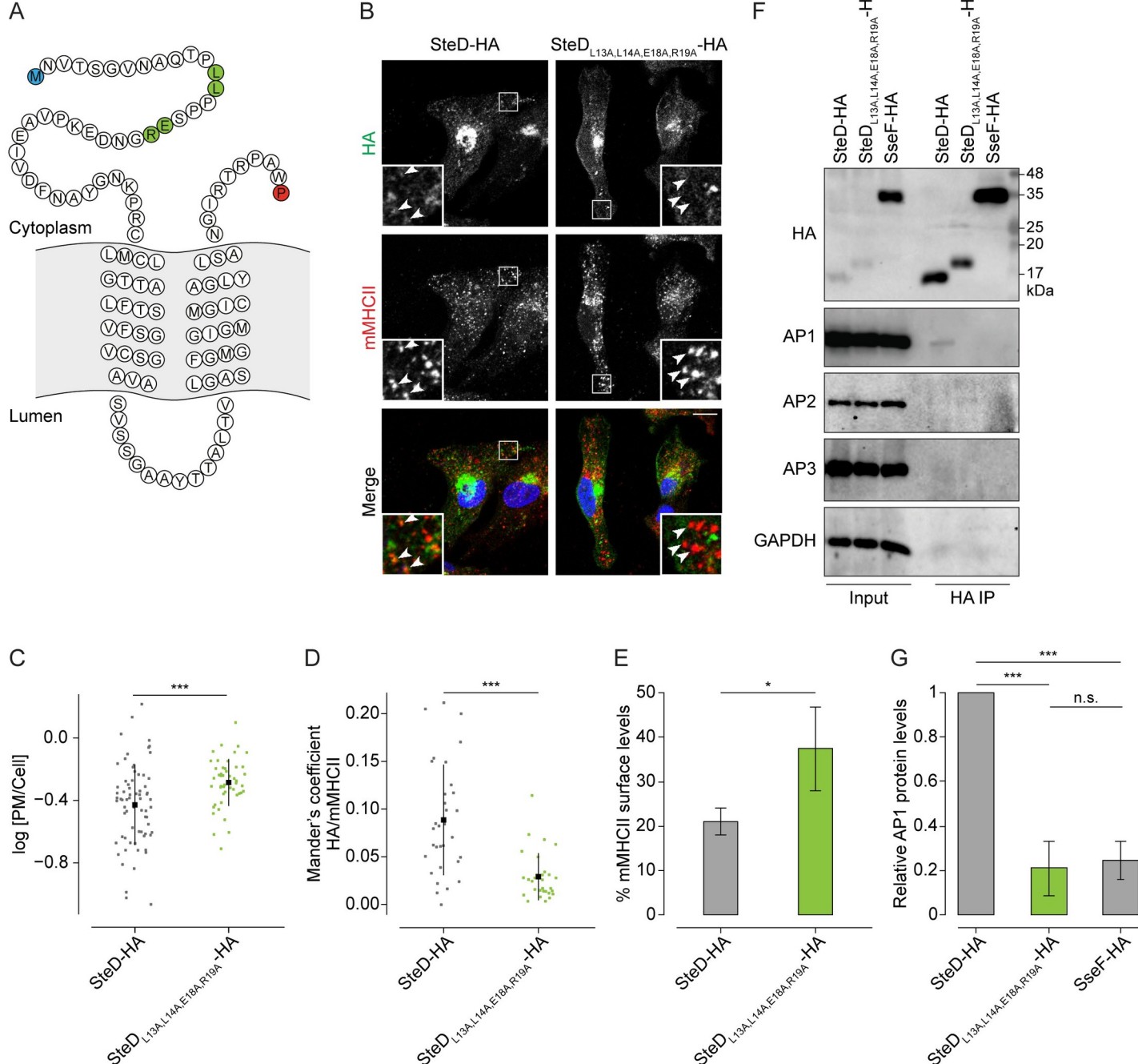

**Fig 7. A dileucine motif-like sequence in the N-terminal tail of SteD mediates post-TGN trafficking.** (A) Amino acid sequence of SteD showing predicted transmembrane domains. The N- and C-terminal residues are highlighted in blue and red respectively. The residues substituted to alanines in SteD$_{L13A,L14A,E18A,R19A}$ are highlighted in green. (B) Representative confocal immunofluorescence microscopy images of Mel Juso cells infected with *ΔsteD Salmonella* strains carrying a plasmid expressing SteD-HA (wt or mutant). Cells were fixed and processed for immunofluorescence microscopy by labelling for HA (green), MHCII compartments (mMHCII, red), and DNA (DAPI, blue). Arrowheads indicate MHCII compartments. Scale bar– 10 μm. (C) Quantification of HA signal at the surface of cells represented in Fig 7B. The fluorescence intensity of the HA signal at the cell surface was measured in relation to total cellular fluorescence. Data are representative of three independent experiments. Each dot represents the value for one cell. Mean ± SD. The log$_{10}$ fold change of the data were analysed by t-test, *** p<0.001. (D) Mander's overlap coefficient of the fraction of SteD-HA positive pixels that colocalise with mMHCII positive pixels from cells represented in Fig 7B. Data are representative of three independent experiments. Each dot represents the value for one cell. Mean ± SD. Data were analysed by t-test, *** p<0.001. (E) mMHCII surface of Mel Juso cells infected with *ΔsteD Salmonella* carrying a plasmid expressing SteD-HA (wt or mutant). Cells were analysed by flow cytometry and amounts of surface mMHCII in infected cells are expressed as a percentage of uninfected cells in the same sample. Mean of three independent experiments done in duplicate ± SD. Data were analysed by paired t-test, * p<0.05. (F) Protein immunoblots of whole-cell lysates (Input) and immunoprecipitation with HA beads (HA IP) from Mel Juso cells

infected with Δ*steD Salmonella* strains carrying a plasmid expressing SteD-HA (wt or mutant) or SseF-HA following crosslinking with DSP. Mutation of charged residues might explain the difference in migration through the SDS gel. AP1 –antibody specific for the γ subunit, AP2 –antibody specific for the α subunit, AP3 – antibody specific for the δ subunit. (G) Levels of immunoprecipitated AP1 were calculated by densitometry from immunoblots as represented in Fig 7F. Protein levels were normalised to wt SteD-HA. Mean of three independent experiments ± SD. The data were analysed by one sample t-test, *** p<0.001, n.s.–not significant.

of GFP-SteD$_{37-111}$. Furthermore, mMHCII surface levels were significantly higher after translocation of SteD$_{L13A,L14A,E18A,R19A}$-HA compared to wt SteD-HA (Fig 7E). Mis-localisation of SteD$_{L13A,L14A,E18A,R19A}$ did not prevent interaction with TMEM127 (S7F and S7G Fig). This suggests that SteD interacts with TMEM127 at the ER, Golgi or TGN through which TMEM127 passes [17], and is consistent with other work showing that SteD can interact with TMEM127 in the absence of mMHCII [6]. Finally, interaction between SteD$_{L13A,L14A,E18A,R19A}$ and AP1 was reduced significantly after translocation from *Salmonella* (Fig 7F and 7G) and ectopic expression (S7F and S7G Fig). Therefore, these residues are required for interaction with AP1, resulting in trafficking of SteD from the TGN to MHCII compartments, which is required for SteD function.

## Discussion

Many bacterial type III secretion system effectors are specifically targeted to organelles and membranes within the infected cell and interference with localisation processes frequently results in their loss of function [18–21]. The targeting mechanisms of transmembrane effectors, which post-translationally integrate into specific host cell membranes, are not well understood. The majority of translocated SteD accumulates at the TGN but its substrates including mature MHCII are located in endosomal compartments and at the PM, raising the question of how SteD reaches these sites. In this work we used site directed mutagenesis of SteD to identify distinct regions of SteD that are required for initial targeting to the ER/Golgi, membrane integration and for AP1-mediated trafficking to MHCII compartments (Fig 8). This demonstrates how a bacterial virulence protein can enter the membrane network of the eukaryotic cell and hijack a vesicular trafficking pathway to regulate its localisation and ultimately its function.

Apart from SteD, a few other effectors have been reported to undergo integration into membranes of the secretory pathway. Tir, a conserved T3SS effector of pathogenic *E. coli* with a similar membrane topology to SteD, localises to the PM where it induces actin pedestals and enables tight attachment of extracellular bacteria by binding to the bacterial surface protein, Intimin [22]. Enterohemorrhagic *E. coli* (EHEC) Tir was also shown to localise at the Golgi network by immunofluorescence and immunoelectron microscopy [23]. Exposure of host cells to BFA prior to infection with EHEC prevented pedestal formation, suggesting that Tir, like SteD must first pass through the Golgi before reaching its site of action at the PM [23]. NleA/EspI, another EHEC effector, contains two putative transmembrane domains and was shown by triton-dependent solubilisation following fractionation of infected host cells to integrate into membranes [24]. NleA/EspI localises to the Golgi through interaction with the COPII component, Sec24 [25]. This interaction stabilises COPII at the Golgi leading to a reduction in general protein secretion [26]. Interestingly, overexpressed NleA interferes with MHCII invariant chain transport and it might thereby also affect antigen presentation [27].

Transmembrane effectors have been suggested to integrate into their target membranes either indirectly, by lateral transfer during translocation into the membrane containing the T3SS translocon, followed by membrane fission and vesicular trafficking to another destination, or directly, following translocation of the effector into the cytoplasm [3]. In the case of SteD, indirect targeting could result from vesicles containing SteD trafficking from the SCV to

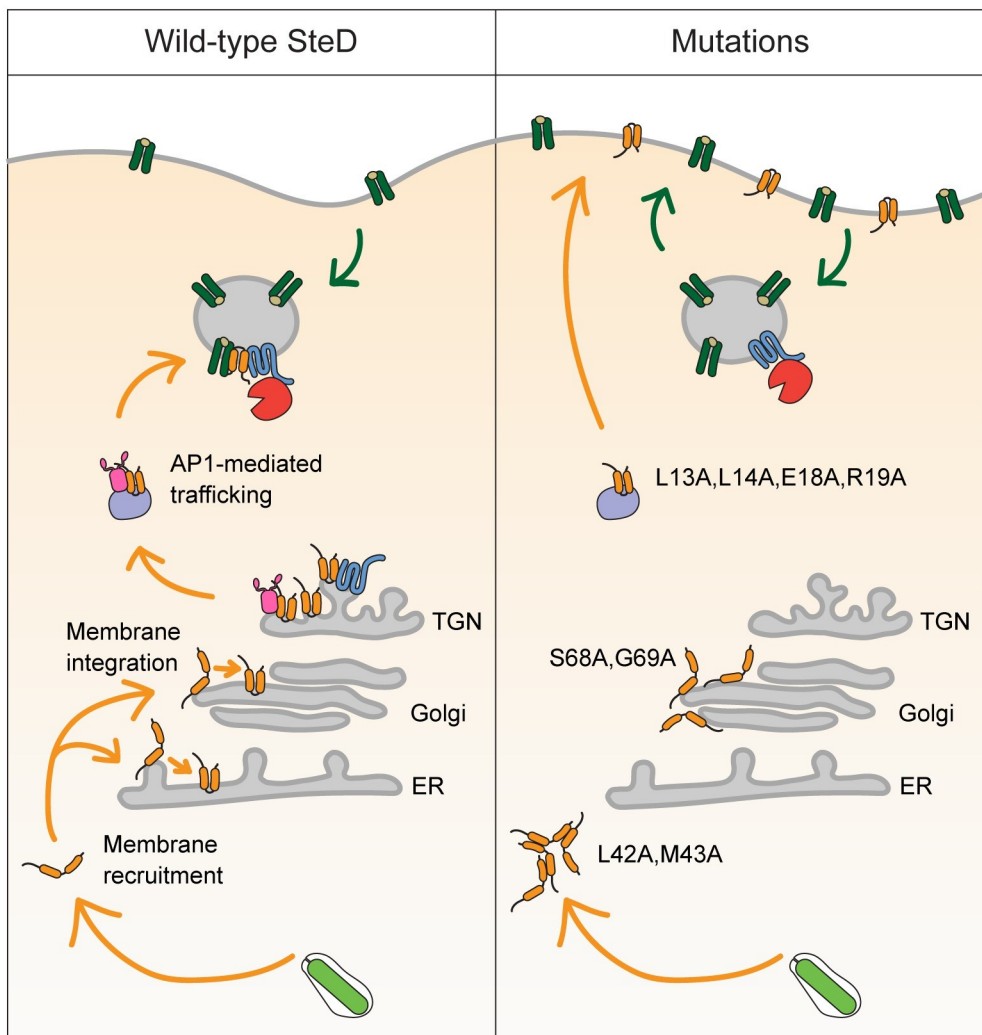

**Fig 8. Model of SteD membrane integration and localisation.** Following translocation from Salmonella (green) into the cytoplasm SteD (orange) is recruited to the membranes of the early secretory pathway where it integrates. SteD then migrates to and accumulates within the TGN. Through interaction with the AP1 complex (pink) it is trafficked to MHCII compartments, where it interacts with mMHCII (dark green), which is ubiquitinated through the actions of TMEM127 (blue) and WWP2 (red) causing a reduction in mMHCII surface levels. SteD$_{L42A,M43A}$ mutation prevents membrane recruitment leading to aggregation in the cytoplasm. SteD$_{S68A,G69A}$ mutation prevents membrane integration resulting in a Golgi-associated non-integrative state. SteD$_{L13A,L14A,E18A,R19A}$ mutation prevents AP1 interaction leading to mis-trafficking of SteD to the plasma membrane.

the Golgi. However, direct integration is more likely for three reasons. First, ectopically expressed GFP-SteD mimics the bacterial translocated effector with respect to Golgi localisation, membrane integration and action on mMHCII [4], (and this work), showing that an SCV membrane and translocon are not required for these processes. Second, Golgi localisation is independent from membrane integration, as demonstrated by SteD mutants incapable of integration, showing that this localisation does not require vesicular trafficking. Third, mutants incapable of either membrane recruitment or integration accumulated within the host cell cytoplasm following translocation.

We propose that SteD is recruited to its target membrane from the cytoplasm through interaction of Region 9 with a protein or lipid, or a combination of components at the

cytoplasmic face of the ER and/or the Golgi cisternae. Identification of the host interaction partner(s) and its involvement in membrane integration is needed to better define the process of SteD integration. Following recruitment, SteD undergoes membrane integration by a mechanism involving Region 13, and specifically S68 and G69. The ability of Region 13 to mediate integration of the transmembrane regions of SseG shows that this mechanism is non-specific. As glycine residues have a high propensity to induce a turn between two transmembrane helices [28], G69 could enable the formation of a hairpin-like structure and hence facilitate the conformation required for integration upon interaction with the membrane. A two-step membrane integration process is consistent with the mechanism of integration proposed for Tir, whereby a region close to the transmembrane domain binds peripherally to the membrane, resulting in a conformational change enabling the hydrophobic domains to adopt an orientation that drives membrane insertion [29]. Insertion of purified Tir into reconstituted membrane vesicles does not require host proteins but does depend on sphingomyelin [29], which is made at the ER and Golgi and enriched at the PM [30].

It is possible that aggregation of $SteD_{ala9}$ and $SteD_{ala13}$ is due to misfolding and that $SteD_{ala13}$ forms aggregates within the membrane. However, both mutants were sensitive to proteasomal degradation, which occurs in the cytoplasm. Membrane proteins must be retro-translocated into the cytoplasm before degradation [31], so their protein levels are unlikely to be sensitive to MG132 treatment. In addition, $SteD_{L42A,M43A}$ and $SteD_{S68A,G69A}$, which fully recapitulated the phenotypes of the full mutants were competent for translocation across the bacterial envelope and vacuole membrane without aggregation. Therefore, the simplest explanation is that the prolonged presence of the exposed hydrophobic transmembrane domains in the host cell cytoplasm causes aggregation and triggers a quality control pathway, leading to ubiquitination of both mutants and their subsequent degradation by the proteasome [32,33].

Microscopic analysis of the non-membrane integrating $SteD_{ala13}$ mutant showed that it remained confined to the Golgi region, indicating that SteD must be in its membrane-integrated state to undergo onward post-TGN transport. SteD thus resembles transmembrane cargo proteins that are recognised for vesicular traffic by adaptor proteins via sorting signals in their cytoplasmic tails, such as the dileucine motif [DE]XXXL[LI]. It is therefore interesting that the LLPPSER sequence required for transport of SteD to MHCII compartments resembles an inverted dileucine motif. This could represent an example of effector mimicry by convergent evolution. However, individual substitution of either leucine inactivates mammalian [DE]XXXL[LI] sorting signals [12], whereas the localisation of the $SteD_{L13A,L14A}$-HA double mutant was similar to that of wt SteD. Furthermore, although inverted leucine-containing motifs can mediate substrate recognition in other proteins [34] it is not clear whether SteD's LLPPSER sequence interacts directly with the AP1 binding site or whether other proteins are involved in the process. Further biochemical and structural studies are required to determine the molecular details of the SteD/AP1 interface.

Other intracellular pathogen proteins have been shown to interact with adaptor proteins through canonical dileucine motifs. The Coxiella burnetii T4SS effector CvpA interacts with AP2 via three dileucine motifs (EESKLL, RHINLL and EIQQLL) and re-routes endocytic compartments to the pathogen-containing vacuole [35]. HIV-1 proteins Nef and Vpu interact with host adaptor protein complexes via ENTSLL and ELSALV sequences respectively, resulting in the redistribution of cell surface proteins involved in cellular immunity [36,37]. By re-routing protein cargo or trafficking pathways these proteins aid pathogen replication. On the other hand, by localising within membranes of the secretory pathway and by virtue of its LLPPSER post-TGN sorting sequence, SteD resembles classical transmembrane protein cargo.

Through interaction with AP1, SteD exploits an established host cell trafficking pathway between the TGN and endosomes [15,38]. Indeed, AP1 mediates delivery of invariant chain-

bound MHCII complexes from the TGN to MHCII compartments [39]. Although blocking MHCII compartment localisation by expressing SteD after Golgi disruption totally prevented SteD function, disrupting its post-TGN traffic through mutation of the LLPPSER sequence or AP1 knockdown only partially disrupted or did not affect SteD function, respectively. It is possible that a small but functionally significant amount of SteD is still able to reach MHCII compartments in these conditions. Alternatively, in the case of SteD$_{L13A,L14A,E18A,R19A}$, this could be due to the partial functionality of SteD at the PM, where it might also come into contact with mMHCII. The differences between AP1 knockdown and mutation of the dileucine sequence in relation to SteD localisation and functionality could be due to incomplete AP1 knockdown or to a redundancy in interaction of wt SteD with other adaptors.

The dependence on AP1 for trafficking to MHCII compartments might well create a rate-limiting step for post TGN SteD traffic, explaining the accumulation of SteD at the TGN. This might also provide a source of SteD to replenish that which is lost by degradation as a result of ubiquitination by the TMEM127/WWP2 machinery [6].

The mutational dissection of SteD described in this and our previous work enables the following functional regions of the protein to be defined: L13, L14, E18 and R19 –an inverted dileucine motif-like sequence involved in post TGN transport to MHCII compartments (this work); K24 –undergoes ubiquitination which contributes to the ability of SteD to induce ubiquitination of mMHCII [6]; L42 and M43 –recruitment to the ER/Golgi (this work); S68 and G69 –integration into membranes (this work); transmembrane regions–intramembrane interaction with TMEM127 [6]; residues in the C-terminal tail–interaction with MHCII [4]. Taken together, this reveals a remarkable level of functional complexity within this small bacterial protein, resulting in the disruption of the adaptive immune response by reducing surface levels of at least three key proteins, CD97 [5], CD86/B7.2 [4] and mMHCII [4].

## Materials and methods

### Bacterial strains, plasmids and antibodies

*Salmonella enterica* serovar Typhimurium (14028s) wild-type and all mutant strains are listed in S1 Table. Bacteria were grown in Luria–Bertani (LB) medium supplemented with carbenicillin (50 μg ml$^{-1}$) or kanamycin (50 μg ml$^{-1}$) as appropriate. All plasmids used are listed in S2 Table. All antibodies used are listed in S3 Table.

### Plasmid construction

All primers used are listed in S4 Table.

Mutations were created using overlap-PCR [40] with specific mutagenesis primers as indicated in S4 Table. The sequence of SseG was amplified from p*myc*::*sseG* [9] and inserted into m4p plasmid with an N-terminal mCherry tag [41] using PciI and NotI. The sequences of GFP-SteD and GFP-SteD$_{ala13}$ were amplified from m4p GFP-SteD and m4p GFP-SteD$_{ala13}$ and inserted into pcDNA 4/TO (Life Technologies) using BamHI and EcoRI. Truncation fragments of SteD were amplified from m4p GFP-SteD with relevant primers and inserted into m4p GFP using PciI and NotI. The sequence of mEos3.2 was amplified from Addgene 57484 and inserted into m4p GFP-SteD and GFP-SteD$_{37-111}$ using NcoI and PciI. All plasmids were checked by sequencing.

### Cell culture and infection

Human Mel Juso cells were maintained in Dulbecco's Modified Eagle Medium (DMEM, Sigma) containing 10% heat-inactivated fetal calf serum (FCS, Gibco) at 37°C in 5% $CO_2$.

When indicated, cells were incubated with DMEM containing MG132 (10 μM, Sigma) or DMSO as a vehicle control (1:1000) for 5 h. To disrupt the Golgi, cells were incubated in DMEM containing BFA (10 μg ml$^{-1}$, Sigma) for 3 h.

Mel Juso cells were infected for 30 min at MOI of 100 with late log-phase *Salmonella* grown in LB. Cells were washed twice with PBS and incubated in fresh medium containing gentamicin (100 μg ml$^{-1}$) for 1 h to kill extracellular bacteria. After 1 h, the antibiotic concentration was reduced to 20 μg ml$^{-1}$, and the cells were processed 20 h post-invasion (p.i.).

For analysis of translocated effectors, cells were lysed in 0.1% Triton X-100 and incubated on ice for 15 min with vortexing. The post-nuclear supernatant (PNS) was separated from the nuclear pellet and non-lysed *Salmonella* cells by centrifugation.

## Transfection

For transient plasmid transfections, plasmids and lipofectamine 2000 were combined and incubated in OptiMEM for 5 min at room temperature before being added to cells. Cells were analysed 16–20 h after plasmid transfection. For siRNA transfections, siRNA and Lipofecta-mine RNAiMAX (Life Technologies) were combined and incubated in OptiMEM for 5 min at room temperature before being added to cells. AP1B1 siRNA mix (UAGACGAGCUUAUCU GCUA, CCACUCAGGACUCAGAUAA, GGAAGGCUGUGCGUGCUAU, CUAAGGACU UGGACUACUA), AP2M1 siRNA mix (GUUAAGCGGUCCAACAUUU, GCGAGAGGGU AUCAAGUAU, AGUUUGAGCUUAUGAGGUA, GAACCGAAGCUGAACUACA), AP3D1 siRNA mix (CUACAGGGCUCUGGAUAUU, GGACGAGGCAAAAUACAUA, GAAG GACGUUCCCAUGGUA, CAAAGUCGAUGGCAUUCGG) and Scrambled siRNA mix (UGGUUUACAUGUCGACUAA, UGGUUUACAUGUUGUGUGA, UGGUUUACAU GUUUUCUGA, UGGUUUACAUGUUUUCCUA) were purchased from Dharmacon and used at 5 pmol. Cells were diluted 24 h after siRNA transfection and analysed 3 days after siRNA transfection.

## Induced expression of SteD

To regulate GFP-SteD expression, a Tet-on system was used. Mel Juso cells stably expressing the Tet Repressor from the pcDNA 6/TR vector (Life Technologies) were made following the manufacturer's instructions. The vector was linearised and transfected into Mel Juso cells as described above. Expressing cells were selected with 10 μg ml$^{-1}$ Blasticidin. A clonal population was selected based on maximum repressor expression. GFP-SteD was cloned into the pcDNA 4/TO vector (Life Technologies) following the manufacturer's instructions (S2 Table) and transiently transfected into the repressor-expressing cells. Expression was induced with DMEM containing 1 μg ml$^{-1}$ doxycycline for 4 h.

## Flow cytometry

Surface levels of mMHCII were measured following infection or transfection of Mel Juso cells as described previously [4] with minor modifications. In brief, Mel Juso cells were detached using 2 mM EDTA in PBS. All antibodies were diluted in FACS buffer (5% FCS and 1 mM EDTA in PBS). See S3 Table for information on primary antibodies; secondary antibodies were purchased from Life Technologies, UK. Cells were labelled with mouse anti-HLA-DR (mMHCII) at 1:300 for 30 min on ice, washed in cold PBS, then labelled with Alexa Fluor 647 donkey anti-mouse at 1:300 for 30 min on ice. For detection of intracellular *Salmonella* and translocated HA-tagged SteD, cells were fixed in 3.7% paraformaldehyde for 1 h at room temperature and permeabilised with 0.1% Triton X-100 in FACS buffer for 10 min at room temperature. Subsequently, cells were labelled with goat anti-*Salmonella* CSA-1 at 1:500 and rat

anti-HA at 1:200 antibodies for 30 min on ice. Cells were washed in cold PBS, then labelled with Alexa Fluor 555 donkey anti-goat and Alexa Fluor 488 donkey anti-rat antibodies both at 1:300 for 30 min on ice. Surface levels of mMHCII were calculated as geometric mean of infected cells or GFP-positive cells/geometric mean of uninfected cells or GFP-negative cells x 100. Calculating the % mMHCII surface levels as a percentage of uninfected or GFP-negative cells provides an internal control for antibody labelling. Data were acquired using Calibur or Fortessa flow cytometers (BD Biosciences) and analysed using FlowJo v10 software.

## Membrane fractionation

Mel Juso cells expressing GFP-tagged SteD variants or infected with *Salmonella* expressing HA-tagged SteD variants were collected and lysed in homogenization buffer (250 mM sucrose, 3 mM imidazole (pH 7.4), and 1 mM phenylmethylsulfonyl fluoride (PMSF)) by mechanical disruption using a Dounce homogenizer. The post-nuclear supernatant was collected after centrifugation at 1,800 $g$ for 15 min and split into three samples. The membrane fraction was pelleted and separated from the soluble fraction in each sample by centrifugation at 100,000 $g$ for 1 h at 4˚C. One membrane pellet was used as the total membrane sample. To remove peripherally-associated proteins, the second membrane pellet was resuspended in 2.5 M urea and incubated for 15 min on ice followed by centrifugation at 100,000 $g$ for 1 h at 4˚C. This yielded a supernatant containing peripherally-associated membrane proteins. To solubilise integral membrane proteins the third membrane pellet was resuspended in RIPA buffer (150 mM NaCl, 1% Triton X-100, 0.5% sodium doxycholate, 0.1% SDS and 50 mM Tris-Cl (pH 8.0)) and incubated on ice for 15 min with vortexing followed by centrifugation at 100,000 $g$ for 1 h at 4˚C. This yielded a pellet containing protein aggregates and a supernatant containing integral membrane proteins. All samples were analysed by SDS PAGE and immunoblotting. The MHCII α chain was used as an integral membrane protein control. Actin was used as a soluble protein control. Golgin-97 was used as a peripherally-associated membrane protein control.

## Immunofluorescence microscopy

Cells were seeded onto coverslips and infected or transfected as described above. Cells were washed in PBS, fixed in 3% paraformaldehyde in PBS for 15 min at room temperature, then the paraformaldehyde was quenched by incubation with 50 mM $NH_4Cl$ for 10 min. All antibodies were diluted in 10% horse serum (Sigma) and 0.1% saponin (Sigma) in PBS. Coverslips were washed in 0.1% saponin in PBS then incubated with appropriate primary antibodies for 1 h at room temperature, washed in 0.1% saponin in PBS, then incubated with secondary antibodies for 1 h at room temperature. Finally, coverslips were incubated with 0.5 μg ml$^{-1}$ DAPI (Invitrogen) for 5 min, washed in 0.1% saponin in PBS then mounted onto glass slides using Aqua-Poly/Mount (Polysciences, Inc.). See S3 Table for information on primary antibodies and dilutions used. Secondary antibodies were purchased from Life Technologies, UK.

## Confocal microscopy and live-cell imaging

All coverslips were imaged at room temperature using a confocal laser scanning microscope (LSM 710, Carl Zeiss) equipped with a Plan Apochromat 63x (Carl Zeiss) oil-immersion objective. For live imaging, cells were seeded in dishes (Matek) with an embedded glass cover slip. Prior to imaging, DMEM was replaced with FluoroBrite (Gibco) containing 10% FCS (Gibco), 40 mM Hepes (Sigma) and 2 mM L-Glutamine (Sigma). Live cells were maintained at 37˚C in a heated chamber. Protein expression was blocked with cycloheximide (50 μg ml$^{-1}$) for 1 h before photobleaching. Photobleaching of GFP was performed using a 488 nm laser. Photoconversion of mEos was performed using a 405 nm laser.

## Image analysis

Quantitative analyses of SteD at the TGN or plasma membrane were done using CellProfiler software [42]. Nuclei were segmented using the DAPI signal. TGN objects were segmented using TGN46 labelling, and the outline of cells were segmented using background labelling from TGN46. Tertiary plasma membrane objects were segmented by expanding and shrinking the cell object by 4 pixels (corresponding to 0.5 μm) and then subtracting the shrunken cell from the expanded cell. Fluorescence intensity measurements were made after extracellular background subtraction using a rolling ball radius of 200 pixels (corresponding to 26.4 μm). Non-transfected or uninfected cells were excluded from the analysis based on a threshold. The mean intensity of the SteD signal was measured from TGN and plasma membrane segmentation masks. The $\log_{10}$ of the ratio of the segmented signal over the total cellular signal was calculated for each cell.

Pearson's correlation coefficient was calculated using ImageJ software. The extracellular background was subtracted from images using the Background Subtraction function in ImageJ, with a rolling ball radius equal to 200 pixels or 26.4 μm. Pearson's correlation coefficient values were obtained from individual cells using the Coloc 2 ImageJ plugin (http://imagej.net/Coloc_2). Mander's overlap coefficient was calculated using ImageJ software to measure the proportion of colocalising pixels between two punctate signals. The extracellular background was subtracted from images as above. Local background was corrected by subtracting the median intensity of a 10 x 10 pixel region surrounding each pixel. Non-specific fluorescence was then subtracted using values measured from unlabelled cells. The images were then converted to binary and the Mander's overlap coefficient was measured from individual cells using the Coloc 2 imageJ plugin.

## Immunoprecipitation

Mel Juso cells expressing GFP-tagged SteD variants or infected with *Salmonella* expressing HA-tagged SteD variants as indicated were harvested in cold PBS, washed and then resuspended in 2 mM dithiobis(succinimidyl propionate) (DSP) (Sigma) and incubated for 2 h at 4˚C to crosslink intracellular proteins before cell lysis. Cells were pelleted and resuspended in 20 mM Tris (pH 8.0) for 15 min at room temperature to stop the crosslinking reaction. Cells were lysed in lysis buffer (5% glycerol, 0.5% Triton X-100, 1 mM PMSF in PBS) for 30 min at 4˚C. The post-nuclear supernatant was obtained by centrifugation at 16,000 *g* for 10 min. Proteins were immunoprecipitated by incubation with anti-HA sepharose beads (Pierce) or anti-GFP-Trap beads (ChromoTek) for 2 h at 4˚C. Immunoprecipitates were washed four times with lysis buffer and boiled in SDS buffer containing 2.5% β-mercaptoethanol and 400 mM DTT before analysis by SDS-PAGE and immunoblotting. Densitometry measurements were carried out using Image Lab software (Bio-Rad).

## Supporting information

**S1 Fig.** (A) Amino acid sequence of SteD showing regions of amino acids substituted to alanine in alanine scanning mutagenesis. (B) Representative flow cytometry plots showing the gating strategy for GFP-positive cells and negative cells as used for Fig 1C. (C) mMHCII surface levels of Mel JuSo cells expressing GFP or GFP-SteD and treated with DMSO or MG132 were measured by flow cytometry. Mean of three independent experiments done in duplicate ± SD. Data were analysed by paired t-test, ** $p < 0.01$, n.s.–not significant. (PDF)

**S2 Fig.** (A) Representative confocal immunofluorescence microscopy images of Mel JuSo cells expressing GFP-SteD (wt or mutants) after MG132 treatment. Cells were fixed and processed for immunofluorescence microscopy by labelling for ubiquitin (UB, red), and DNA (DAPI, blue). Arrowheads indicate cellular aggregates. Scale bar– 10 μm.
(PDF)

**S3 Fig.** (A and D) Protein immunoblots of whole-cell lysates (WCL) and post-nuclear super-natant (PNS) of Mel Juso cells infected with *ΔsteD Salmonella* strains carrying a plasmid expressing SteD-HA (wt or mutants) and treated with MG132 or DMSO carrier. Actin and DnaK represent host cell and *Salmonella* loading controls respectively. (B and C) mMHCII surface levels of Mel JuSo cells expressing GFP or GFP-SteD (wt or mutants) and treated with MG132. Cells were analysed by flow cytometry and amounts of surface mMHCII in GFP-positive cells are expressed as a percentage of GFP-negative cells in the same sample. Mean of three independent experiments done in duplicate ± SD. Data were analysed by one-way ANOVA followed by Dunnett's multiple comparison test in comparison to wt SteD, *** $p<0.001$, ** $p<0.01$, * $p<0.05$, n.s.–not significant. (E) Representative flow cytometry plots showing the gating strategy for HA-positive and negative cells as used for Fig 3B.
(PDF)

**S4 Fig.** (A) Representative confocal immunofluorescence microscopy images of Mel JuSo cells expressing GFP-SteD under a doxycycline-regulated promoter. Cells were either treated with doxycycline for 4 h (dox) or treated with BFA for 3 h followed by doxycycline and BFA for 4 h (BFA-dox). Cells were then fixed and processed for immunofluorescence microscopy by label-ling for MHCII compartments (mMHCII, red), the TGN (TGN46, grey), and DNA (DAPI, blue). Arrowheads indicate MHCII compartments. Scale bar– 10 μm. (B) Representative flow cytometry plots showing the gating strategy for GFP-positive and negative cells as used for Fig 4D. (C) Confocal microscopy images demonstrating photobleaching of a Mel JuSo cell expressing GFP-SteD from S2 Video. Closed arrowheads indicate anterograde vesicle traffic from the Golgi region. Barbed arrowhead indicates retrograde vesicle traffic. Scale bar– 10 μm.
(PDF)

**S5 Fig.** (A) mMHCII surface levels of Mel JuSo cells expressing GFP-SteD and treated with scrambled siRNA (SCR) or siRNA specific to the β subunit of AP1. Cells were analysed by flow cytometry and amounts of surface mMHCII in GFP-positive cells are expressed as a percent-age of GFP-negative cells in the same sample. Mean of three independent experiments done in duplicate ± SD. Data were analysed by one-way ANOVA followed by Dunnett's multiple com-parison test n.s.–not significant. (B) Quantification of GFP at the surface of cells represented in Fig 5D. The fluorescence intensity of the GFP signal at the surface of cells was measured in relation to total cellular fluorescence. Data are representative of three independent experi-ments. Each dot represents the value for one cell. Mean ± SD. The $\log_{10}$ fold change of the data were analysed by t-test, n.s.–not significant.
(PDF)

**S6 Fig.** (A) Protein immunoblots of Mel JuSo cells expressing GFP or GFP-SteD (wt or mutants). (B) Quantification of GFP at the TGN of cells represented in Fig 6C. The fluores-cence intensity of the GFP signal at the TGN was measured in relation to total cellular fluores-cence. Data are representative of three independent experiments. Each dot represents the value for one cell. Mean ± SD. The $\log_{10}$ fold change of the data were analysed by one-way ANOVA followed by Dunnett's multiple comparison test, *** $p<0.001$, n.s.–not significant. (C) Confocal microscopy images demonstrating photoactivation of a Mel JuSo cell expressing mEos-SteD (wt or 37–111) from S3 Video. Red dotted circles indicate photo-activated areas.

Red arrowheads indicate Golgi-derived vesicles. Scale bar– 10 μm.
(PDF)

**S7 Fig.** (A) mMHCII surface levels of Mel JuSo cells expressing GFP or GFP-SteD (wt or mutants). Cells were analysed by flow cytometry and amounts of surface mMHCII in GFP-positive cells are expressed as a percentage of GFP negative cells in the same sample. Mean of three independent experiments done in duplicate ± SD. Data were analysed by one-way ANOVA followed by Dunnett's multiple comparison test compared to wt SteD, *** p<0.001, ** p<0.01, n.s.–not significant. (B) mMHCII surface levels of Mel Juso cells infected with *ΔsteD Salmonella* carrying a plasmid expressing SteD-HA (wt or mutant). Cells were analysed by flow cytometry and amounts of surface mMHCII in infected cells are expressed as a percentage of uninfected cells in the same sample. Mean of three independent experiments done in duplicate ± SD. Data were analysed by one-way ANOVA followed by Dunnett's multiple comparison test, *** p<0.001, n.s.–not significant. (C) Representative confocal immunofluorescence microscopy images of Mel JuSo cells expressing GFP-SteD (wt or mutants). Cells were fixed and processed for immunofluorescence microscopy by labelling for MHCII compartments (mMHCII, red), the TGN (TGN46, grey), and DNA (DAPI, blue). Arrowheads indicate MHCII compartments. Scale bar– 10 μm. (D) Quantification of GFP at the surface of cells represented in S7C Fig. The fluorescence intensity of the GFP signal at the cell surface was measured in relation to total cellular fluorescence. Data are representative of three independent experiments. Each dot represents the value for one cell. Mean ± SD. The $log_{10}$ fold change of the data were analysed by one-way ANOVA followed by Dunnett's multiple comparison test, *** p<0.001, n.s.–not significant. (E) Mander's overlap coefficient of the fraction of GFP-SteD positive pixels that colocalise with mMHCII positive pixels from cells as represented in S7C Fig. Data are representative of three independent experiments. Each dot represents the value for one cell. Mean ± SD. Data were analysed by one-way ANOVA followed by Dunnett's multiple comparison test, *** p<0.001, n.s.–not significant. (F) Protein immunoblots of whole-cell lysates (Input) and immunoprecipitation with GFP-trap beads (GFP IP) from Mel Juso cells expressing GFP-SteD (wt or mutants) or GFP-SseG following crosslinking with DSP. Mutation of charged residues might explain the difference in migration through the SDS gel. AP1 –antibody specific for the γ subunit, AP2 –antibody specific for the α subunit, AP3 –antibody specific for the δ subunit. (G) Levels of immunoprecipitated AP1 and TMEM127 were calculated by densitometry from immunoblots as represented in S7F Fig. Protein levels were normalised to GFP-SteD. Mean of three independent experiments ± SD. The data were analysed by one sample t-test, *** p<0.001, * p<0.05, n.s.–not significant. (H) Representative confocal immunofluorescence microscopy images of Mel Juso cells infected with *ΔsteD Salmonella* strains carrying a plasmid expressing SteD-HA (wt or mutant). Cells were fixed and processed for immunofluorescence microscopy by labelling for HA (green), the TGN (TGN46, red), and DNA (DAPI, blue). Scale bar– 10 μm.
(PDF)

**S1 Video. Time-lapse microscopy of Mel JuSo cells expressing GFP-SteD (wt or mutant) treated with MG132.** Scale bar– 10 μm.
(AVI)

**S2 Video. Time-lapse microscopy of Mel JuSo cells expressing GFP-SteD following photobleaching of non-Golgi regions.** Scale bar– 10 μm.
(AVI)

**S3 Video. Time-lapse microscopy of Mel JuSo cells expressing mEos-SteD (wt or mutant) following activation of mEos in Golgi regions.** Scale bar– 10 μm.
(AVI)

**S1 Table. *S.* Typhimurium strains used in this study.**
(PDF)

**S2 Table. Plasmids used in this study.**
(PDF)

**S3 Table. Primary antibodies used in this study.**
(PDF)

**S4 Table. Primers used in this study.**
(PDF)

## Acknowledgments

The authors would like to thank Teresa Thurston and Peter Hill for helpful comments on the manuscript. We are grateful to Jacques Neefjes for providing Mel Juso cells. mCherry-ER-3 was a gift from Michael Davidson (Addgene plasmid 55041), mEos3.2-Tubulin-C-18 was a gift from Michael Davidson (Addgene plasmid 57484).

## Author Contributions

**Conceptualization:** Camilla Godlee, Ondrej Cerny, David W. Holden.

**Funding acquisition:** David W. Holden.

**Investigation:** Camilla Godlee, Ondrej Cerny, Mei Liu, Samkeliso Blundell, Alanna E. Gallagher, Meriam Shahin.

**Supervision:** Camilla Godlee, David W. Holden.

**Writing – original draft:** Camilla Godlee, David W. Holden.

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
