## [Decision Letter · Decision Letter 0]

7 Feb 2022

Dear Dr. Godlee,

Thank you very much for submitting your manuscript "The Salmonella transmembrane effector SteD hijacks AP1-mediated vesicular trafficking for delivery to antigen-loading MHCII compartments" for consideration at PLOS Pathogens. As with all papers reviewed by the journal, your manuscript was reviewed by members of the editorial board and by several independent reviewers. In light of the reviews (below this email), we would like to invite the resubmission of a significantly-revised version that takes into account the reviewers' comments.

The main concern is that misfolding and aggregation of the mutant SteD proteins could account for the defects in membrane localization and function rather than specific functional requirements for region9 and region13. A deeper analysis of the state of the mutants is necessary to prove or disprove membrane integration as described by Reviewer 2.

We cannot make any decision about publication until we have seen the revised manuscript and your response to the reviewers' comments. Your revised manuscript is also likely to be sent to reviewers for further evaluation.

Sincerely,

Denise M. Monack

Section Editor

PLOS Pathogens

Denise Monack

Section Editor

PLOS Pathogens

Kasturi Haldar

Editor-in-Chief

PLOS Pathogens

orcid.org/0000-0001-5065-158X

Michael Malim

Editor-in-Chief

PLOS Pathogens

orcid.org/0000-0002-7699-2064

The main concern is that misfolding and aggregation of the mutant SteD proteins could account for the defects in membrane localization and function rather than specific functional requirements for region9 and region13. A deeper analysis of the state of the mutants is necessary to prove or disprove membrane integration as described by Reviewer 2.

Reviewer's Responses to Questions

**Part I - Summary**

Reviewer #1: Bacterial effector proteins delivered by secretion systems drive host cell modulation processes that enable pathogens’ intracellular virulence traits. While their functions are often characterized, how they reach their intracellular compartment of action once delivered into host cells is rarely understood. In this manuscript, Godlee and colleagues identify the intrinsic features and characterize the intracellular trafficking of the SPI-2 effector SteD that mediate interference with MHCII surface presentation. Using an elegant combination of ectopic expression and infection approaches and well-established assays, the authors use alanine scanning mutagenesis to identify specific regions and motifs of SteD that mediate membrane recruitment and integration in early secretory membranes and uncover a non-canonical sorting signal that allows SteD to traffic via the AP1 adaptor from the TGN to the MHCII-containing compartment.

This is a very well designed and performed study that present convincing data that are rigorously analyzed and support the authors’ conclusions. Although the work focuses on a single effector that is well characterized from a functional standpoint, the findings presented in this study are novel, have broad implications to many bacterial effector functions and are therefore of broad interest to the field.

Reviewer #2: The present study by Godlee at al. investigates the membrane targeting, integration and trafficking of the Salmonella T3SS transmembrane effector SteD, which serves in reducing the expression of MHCII on infected cells. The subject of study is highly relevant as it showcases the deep integration of bacterial pathogenicity into host cell biology on a molecular level. The manuscript is very well written, and the figures are well structured and accessible.

From my perspective, this study falls into two parts: a highly speculative part I that deals with the analysis of membrane targeting and integration of SteD and a very solid part II that describes the the trafficking of SteD from the TGN to the plasma membrane.

In part I, the authors analyze the membrane integration of alanine mutants of SteD. They find that mutation of sequences in the beginning of as well as right behind transmembrane segment 1 (TMS1) result in destabilized, non-functional proteins that cannot be extracted from a membrane fraction, neither with 2.5 M urea nor with RIPA buffer. Surprisingly, the authors conclude that both mutants fail to integrate into membranes. Neither does failed extraction with a low concentration of 2.5 M urea prove membrane integration nor does failed extraction with RIPA buffer disprove membrane integration. If these mutants are truly not membrane integrated, the authors should show this by extracting with 8 M urea instead. Specific interactions of peripheral membrane proteins may be disrupted with 2.5 M urea (as shown for Golgin-97) but aggregates may not be solubilized at this low concentration of urea but at a higher one. Also, intramembrane aggregates exist that resist extraction with fairly mild buffers like RIPA. Typically, intramembrane aggregates are revealed by their extractability using 1% Fos-Choline 12, which could be used by the authors.

The authors conclude that the ala9 mutant (at the N-terminus of the TMS1) clusters in aggregates, which is likely to be correct given its co-localization with ubiquitin. Based on these results the authors hypothesize that the mutated sequence 9 mediates targeting of SteD to the membrane for which I see no evidence, however. Sequence 9 is the major contributor to the overall hydrophobicity of TMS1 (containing two leucines) and mutation to alanines simply reduces the hydrophobicity and may lead to a semi-hydrophobic sequence that can't integrate into membranes but isn't soluble either, so it aggregates. That means, the finding is highly unspecific and does not allow to draw any conclusion towards a specific function of the sequence 9 of SteD.

The authors conclude that the ala13 mutant is unable to integrate into membranes. A brief analysis of the membrane integration propensity of the ala13 mutant tells that it has an even increased hydrophobicity than the wt. This finding, together with the microscopy analysis the authors show throughout the paper, suggests to me that membrane integration of the ala13 mutant is very well possible but that it likely forms intramembrane aggregates. Consequently, I do not buy the main conclusion the authors draw from their findings presented in Figures 1 through 3. Also the statement that the TMS of SseG suffice to localize the protein even in the absence of membrane integration is likely not correct.

In my view, the manuscript gets much better from Figure 4 on, where the authors show that membrane integration of SteD does not occur at the TGN but at the ER, which seems to be similar for the ala13 mutant but unfortunately, the authors do not provide data on the biochemical fractionation of the mutant (Fig. 4C).

From Figure 5 on the authors nicely show the requirements of SteD for sorting from the TGN to MHCII-containing compartments and to the plasma membrane. They show a specific interaction with the adapter protein AP1 and identify the SteD sequence motif that is responsible for AP1-dependent targeting. These data look really conclusive and nicely illustrate the intra host biology of SteD.

I suggest to limit the manuscript to the data investigating the site of SteD integration and the trafficking and to omit the uncertain membrane integration data. The manuscript would much gain from this and shed new light on the intricate integration of bacterial effectors into the host’s cell biology. Interestingly, the title of the manuscript already points in this direction.

Reviewer #3: Godlee et al have investigated the mechanism by which SteD, a Salmonella T3SS effector, traffics to MHC11 compartments. Previously, work from this lab showed that ectopically expressed SteD in antigen presenting cells (Mel Juso), integrates into host membranes and localizes to the TGN and to endosomal compartments including those containing MHCII. Here they have addressed, the mechanisms by which this effector integrates into membranes and localizes to the TGN and endosomes. They started with two alanine substitution mutants, from their previous paper, in each of which a block of 5 amino acids has been substituted with alanines. Ectopic expression of these mutants, SteDala9 and SteDala13, in Mel Juso cells does not result in detectable levels of protein expression (by Western blot) unless proteosome degradation is inhibited in the presence of MG123. Thus, activity and localization of the mutants was analyzed in Mel Juso cells in the presence of MG123. The majority of experiments were carried out using this approach although translocation by Salmonella was demonstrated using double alanine substitutions, SteDL42A,M43A and SteDS68A,G69A, since the original mutants were not stably expressed or translocated by Salmonella. The two translocated mutants recapitulated the results obtained with ectopic expression of SteDala9 and SteDala13. In the second half of the paper they showed that SteD integrates into membranes in the ER or Golgi, then trafficks via the TGN to MHCII compartments. By co-immunoprecipitation they identified AP1 as a specific interactor with SteD. Surprisingly siRNA knockdown of AP1 reduced colocalization of SteD and MHCII yet had no effect on the SteD-dependent reduction of MHCII on the cell surface. Finally, they identify a localization signal, LLPPSER, that resembles an inverted mammalian dileucine motif although mutation of the two lysine residues had no impact on SteD localization.

The paper is very well written and the experiments are well done. The results are significant in that they show novel aspects of T3 effector protein activity and how this can impact the host response, although the later was largely addressed in their previous paper.

**Part II – Major Issues: Key Experiments Required for Acceptance**

Reviewer #1: None

Reviewer #2: As mentioned above, I do have my difficulties with the experiments shown in Fig. 1-3. A deeper analysis of the state of the mutants is necessary to prove or disprove membrane integration as described above.

As it is right now, all statements based on the current findings that relate to a role of sequence 9 in membrane targeting and of sequence 13 in membrane integration should be toned down substantially or better be removed.

Reviewer #3: My main concern is that misfolding and aggregation of the mutant SteD proteins could account for the defects in membrane localization and function rather than specific functional requirements for region9 and region13. However, I am not sure how this can be best addressed since even the mutants with only two alanine substitutions seem to be targeted for degradation.

**Part III – Minor Issues: Editorial and Data Presentation Modifications**

Reviewer #1: 1. My main concern is with the SteD-AP1 interaction, which does not appear very convincing given how little of AP1 is detected in the SteD immunoprecipitates. While the authors use SseG as a negative control for comparison, I am wondering if the fraction of AP1 that interacts with SteD (how much of the total AP1 is it?) provides enough relevance of AP1-dependent transport of SteD from the TGN to endosomes and whether an AP1-independent pathway also operates. This may also explain why the AP1 knockdown has little effect on SteD-mediated decrease in surface MHCII. Could the authors show as comparison how much a known AP1-dependent host protein co-IP AP1? Also, does AP1 label SteD-containing TGN-derived vesicles? Arguably, the loss of colocalization between SteD and MHCII upon AP1 knockdown and the identification of the inverted di-leucine motifs in SteD supports the role of AP1 in SteD traffic, but perhaps this aspect of the study could be strengthened or further discussed.

2. Fig. 1B: the expression levels of SteDala9 and SteDala13 still remain very low upon MG132 treatment. Could this be due to a lack of solubility of SteDala9 and SteDala13 aggregates? How do the authors envision that SteDala13 traffics to Golgi membranes if it is in an insoluble state?

3. Line 235: should be MHCII, not MCHII

4. Fig. S1B: the legend of the flow plot should indicate MHCII, not MHII

Reviewer #2: I miss information on how the used plasmids were created. Plasmid sequences as well as primers and molecular biology operations that would help to re-create them are missing. As it is, it is not possible to know which sequence of SseG was exactly used for the experiments shown in Figure 3 and how the used fusions look like exactly.

Reviewer #3: Figure 4D. The way surface expression of MHCII is shown here is confusing compared to Fig S1C where the geometric mean is shown. Why not show it the same way in both figures?

PLOS authors have the option to publish the peer review history of their article (what does this mean?). If published, this will include your full peer review and any attached files.

Reviewer #1: No

Reviewer #2: No

Reviewer #3: No
---

## [Editor Report · Decision Letter 1]

27 Apr 2022

Dear Dr. Godlee,

We are pleased to inform you that your manuscript 'The Salmonella transmembrane effector SteD hijacks AP1-mediated vesicular trafficking for delivery to antigen-loading MHCII compartments' has been provisionally accepted for publication in PLOS Pathogens.

Best regards,

Denise M. Monack

Section Editor

PLOS Pathogens

Denise Monack

Section Editor

PLOS Pathogens

Kasturi Haldar

Editor-in-Chief

PLOS Pathogens

orcid.org/0000-0001-5065-158X

Michael Malim

Editor-in-Chief

PLOS Pathogens

orcid.org/0000-0002-7699-2064
---

## [Editor Report · Acceptance letter]

24 May 2022

Dear Dr. Godlee,

We are delighted to inform you that your manuscript, "The *Salmonella* transmembrane effector SteD hijacks AP1-mediated vesicular trafficking for delivery to antigen-loading MHCII compartments," has been formally accepted for publication in PLOS Pathogens.

Best regards,

Kasturi Haldar

Editor-in-Chief

PLOS Pathogens

orcid.org/0000-0001-5065-158X

Michael Malim

Editor-in-Chief

PLOS Pathogens

orcid.org/0000-0002-7699-2064